# NSD3 protein methylation and stabilization transforms human ES cells into variant state

Vignesh K Krishnamoorthy[1,2], Fariha Hamdani[1], Pooja Shukla[1], Radhika Arasala Rao[3], Shaikh Anaitullah[3], Kriti Kestur Biligiri[1,2], Rajashekar Varma Kadumuri[4], Purushotham Reddy Pothula[5], Sreenivas Chavali[4], Shravanti Rampalli[1,2]

Cultured human embryonic stem cells (hESCs) can develop genetic anomalies that increase their susceptibility to transformation. In this study, we characterized a variant hESC (vhESC) line and investigated the molecular mechanisms leading to the drift towards a transformed state. Our findings revealed that vhESCs up-regulate EMT-specific markers, accelerate wound healing, exhibit compromised lineage differentiation, and retain pluripotency gene expression in teratomas. Furthermore, we discovered an altered epigenomic landscape and overexpression of the lysine methyltransferases EHMT1, EHMT2, and NSD group of proteins in vhESCs. Remarkably, depleting NSD3 oncogene reversed the molecular and phenotypic changes in vhESCs. We identified a detailed mechanism where EHMT2 interacts and methylates NSD3 at lysine 477, stabilizing its protein levels in vhESCs. In addition, we showed that NSD3 levels are regulated by protein degradation in hESCs, and its stabilization leads to the emergence of the variant state. Overall, our study identify that misregulation of NSD3 in pluripotent stem cells, through methylation-mediated abrogation of its protein degradation, drives hESCs towards oncogenic transformation.

## Introduction

Embryonic stem (ES) cells are pluripotent stem cells derived from the inner cell mass of the embryo. The ability of these cells to maintain self-renewal and/or undergo differentiation in vitro makes them a potential tool for clinical transplantation (1). Studies from several independent laboratories have illustrated frequent alterations in the genome of high passage hES lines (variant hES) (2, 3, 4, 5). These genetic alterations are known to provide selective advantage to hES cells such as greater cell survival and self-renewal properties (6). Such cells are now recognized as transformed or variant human ES cells. Mechanical cues tend to mediate competitive interactions between normal human embryonic stem cells and vhESCs, thereby facilitating rapid expansion of vhESCs in cultures (7). Careful characterization of variant hESCs indicated aberrant differentiation into one or more lineages and acquisition of neoplastic characteristics (8, 9). Some of the genetic alterations seen in vhESCs are commonly observed in cancerous counterparts of ES cells such as embryonic carcinoma cells (8). In 2019, our laboratory reported a variant hES (V-H9-hESC derived from H9-hESC) line with normal chromosome numbers and pluripotency expression profile but has altered traits such as the loose mesenchymal colony morphology, low expression of E-cadherin, higher rates of cell survival, and high expression of the anti-apoptotic gene Bcl-xl (10). However, the detailed molecular events leading to the variant state have not been studied.

Human ES cells remain closely associated with neighbouring stem cells via E-cadherin in compact colonies. The expression of E-cadherin in hESC colonies is considered as characteristic undifferentiated pluripotent stem cells (11, 12, 13). Accordingly, E-cadherin expression is decreased immediately after induction of differentiation (14, 15). E-cadherin–mediated cell–cell interactions play an important role in the survival and self-renewal of hESCs (16). Forced depletion of E-cadherin activity by antibodies or by CRISPR interference system results in loss of cell–cell contact, emergence of mesenchymal-like phenotype via epithelial-to-mesenchymal transition (EMT), and enhanced motility (17). However, interestingly, E-cadherin depleted hESCs retained their pluripotent nature (17, 18, 19). Traits such as pluripotent gene signatures coupled with loss of E-cadherin and EMT phenotype are known hallmarks of malignant cancers (8). Because the characteristics of variant hES derivatives isolated in our laboratory overlap with cancer cells, variants can serve as a powerful tool for delineating the sequential molecular changes in the route of oncogenesis and human disease in laboratory settings (20).

The *E-cadherin* gene is extensively regulated via epigenetic mechanisms. For example, CpG island hypermethylation around the

[1]Council of Scientific and Industrial Research (CSIR) – Institute of Genomics and Integrative Biology (IGIB), New Delhi, India   [2]Academy of Scientific and Innovative Research (AcSIR), Ghaziabad, India   [3]Institute for Stem Cell Science and Regenerative Medicine (DBT-inStem), GKVK Campus, Bangalore, India   [4]Department of Biology, Indian Institute of Science Education and Research (IISER) Tirupati, Tirupati, India   [5]National Centre for Biological Sciences, GKVK Campus, Bangalore, India

Correspondence: shravanti@igib.in; shravanti@igib.res.in

promoter of E-cadherin is linked to loss of gene expression (21, 22). This DNA hypermethylation is accompanied by repressive histone modifications such as a H3K27me3 mark deposited by enhancer of zeste homolog 2 (EZH2) (23, 24, 25, 26). EHMT2, which deposits repressive H3K9me2/3 marks, is also known to regulate *E-cadherin* promoter (27). Furthermore, the overexpression of H3K36 methyltransferase NSD3 induces EMT as evident by loss of E-cadherin and gain of mesenchymal markers such as vimentin and N-cadherin (28). In addition, somatic and germline mutations in *E-cadherin* genes were also reported in breast, gastric, and liver cancers (29, 30, 31). Overall, dysregulation of genetic or epigenetic mechanisms governing the E-cadherin gene has also been implicated in the alteration in pathways regulating cell proliferation, survival, and maintenance of the epithelial state in cancers (32). Although the nature of E-cadherin regulation is extensively studied in cancer biology (33), the mechanisms governing E-cadherin expression and the downstream signalling towards maintenance of the pluripotent epithelial state in human ES cells remain unknown.

In the current study, we set out to investigate the regulation of the epithelial state in human pluripotent stem cells by studying the normal and variant human ES cells. We demonstrate that the V-H9-hESC has acquired the markers related to EMT and has altered propensity of differentiation towards astroglial and cardiac lineages. Teratomas derived from the variant hESC retain the expression of pluripotency markers such as Oct4 and Sox2 indicative of oncogenic propensity. V-H9-hESCs exhibit an accelerated proliferation and overexpression of BCL-XL and BCL2L genes, accompanied by the misexpression of E-cadherin regulating lysine methyltransferases such as EHMT1, EHMT2, and NSD (NSD1-3) proteins. Genetic depletion of NSD3 was sufficient to revert the molecular defects of EMT, accelerated cell proliferation, and defective differentiation in variants. At the mechanistic level, we identified the methylation of NSD3 isoforms (NSD3S and NSD3L) by EHMT2 stabilizes NSD3 protein in V-H9-hESCs. Furthermore, we demonstrate NSD3 (NSD3L and NSD3S) is methylated at histone-like conserved lysine (K477) of the "ARKS" motif. Methylation on NSD3 stabilizes the protein, which is otherwise a short-lived protein. Particularly, the K477 methyl mutant protein of NSD3 is susceptible to degradation and can be rescued upon treatment of the ubiquitin–proteasome inhibitor MG132, indicating the significance of this residue in antagonizing ubiquitin-mediated degradation. Overall, our studies demonstrate that E-cadherin expression is strongly regulated by NSD3 proteostasis in pluripotent cells and the fate of hESCs towards the oncogenic route is triggered by the overexpression of NSD3.

# Results

### Variant hESCs exhibit features of EMT

Pluripotent embryonic stem cells (ES cells) are epithelial in nature and express E-cadherin (E-Cad) robustly (34). Human ES cells that are cultured independent of feeder cells show colonies of undifferentiated compact cells with fibroblast-like mesenchymal cells at the periphery (6). Previously, we reported a V-H9-hESC line, which showed expansion of fibroblast-like cells coupled with loss of E-cadherin (10). Long-term passage of V-H9-hESC colonies leads to homogenized culture of flattened fibroblast-like cells and near-complete loss of compact cells (Fig 1A and B). These cells expressed stemness factors (Oct4 and Nanog) at comparable levels to that of normal H9-hESCs but had reduced c-Myc and E-cadherin transcripts (Fig S1A–D). Lowered E-cadherin mRNA correlated with a negligible amount of protein (Fig 1D). However, c-Myc protein levels were higher and did not correlate with its transcript levels indicating c-Myc might be regulated by post-transcriptional/translational mechanisms (Fig 1C).

In the light of our morphological observations above, we speculated that V-H9-hESCs represent a cell of EMT. The EMT involves a complex network of signalling pathways and action of various transcription factors such as snail, slug, and β-catenin, which leads to down-regulation of E-cadherin and up-regulation of N-cadherin (35, 36, 37). Therefore, we investigated the expression of N-cadherin, snail, and β-catenin in V-H9-hESCs. A switch in cadherin expression was found with a strong positive N-cadherin expression in the mesenchymal-like cells V-H9-hESCs (37, 38) (Fig 1D). β-Catenin, another typical EMT marker, showed reduced expression in V-H9-hESC line (37) (Fig S1E). Vimentin, a type III intermediate filament expressed in mesenchymal cells, facilitates migration of epithelial cells during embryological, organogenetic, or pathological processes (37, 39, 40). To confirm the EMT phenotype, we next performed immunostaining analysis of vimentin in normal and variant hESCs. Intense and homogeneous staining of vimentin was observed throughout the variant hESC colony; on the contrary, low levels of vimentin were seen in the peripheral cells of the H9-hESC colony (Fig 1E). The expression of the nuclear transcription factor snail in human ES cells undergoing EMT is well documented (41). Therefore, we investigated the expression of snail in H9-hESCs and V-H9-hESCs. Low levels of cytoplasmic snail were noticed in H9-hESCs, whereas elevated levels of snail in the nuclear compartment were seen in V-H9-hESCs (Fig 1E). Cumulatively, these observations confirm that V-H9-hESCs indeed represent the EMT phenotype along with the sustained expression of pluripotent transcription factors.

Cells undergoing EMT are promigratory in nature (42); therefore, we examined migration competency of the V-H9-hESC in response to the mechanical scratch wound. Scratch wounds were created in the colonies of H9- and V-H9-hESC, and regrowth into a scratched area was monitored by microscopy over 3 d. In H9-hESCs, cells at the edge migrated as sheet and closed scratch in 72 h (Fig 1F), whereas in V-H9-hESCs, we observed single cells populating the scratch area. Migrated cells grew rapidly and occupied the entire scratched area in 24 h (Fig 1G), and the scratch area became confluent by 36 h. Quantification of the percentage of open wound area clearly demonstrated that V-H9-hESCs possess accelerated cell migration properties compared with H9-hESCs (Fig 1H). Overall, our results demonstrated EMT signature and enhanced migration in V-H9-hESCs.

### Variant hESCs exhibit impaired differentiation

Next, we examined multilineage differentiation potential of V-H9-hESCs using directed differentiation protocols towards neuronal

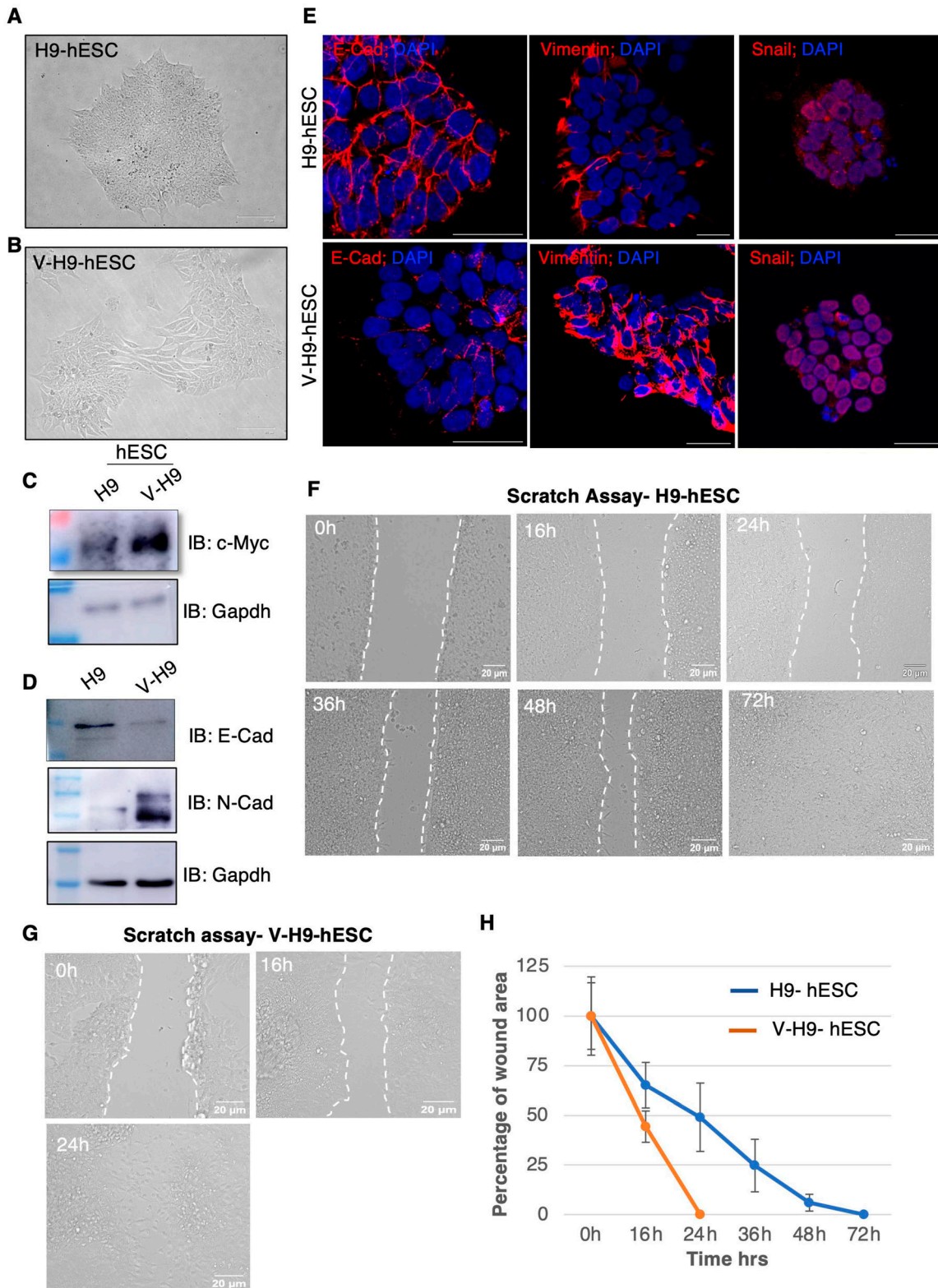

**Figure 1. Detailed characterization of the variant human ES line.**
**(A)** Phase-contrast image of H9-hESCs depicting the morphology of the colonies (scale bar = 125 $\mu$m). **(B)** Phase-contrast image of V-H9-hESCs depicting the morphology of the colonies (scale bar = 125 $\mu$m). **(C, D)** Western blot analysis of c-Myc E-cadherin and N-cadherin in H9-hESCs and V-H9-hESCs. **(E)** Immunostaining for E-cadherin, vimentin, and snail in H9-hESCs and V-H9-hESCs (scale bar = 40 $\mu$m). **(F, G)** Representative bright-field images of H9-hESC and V-H9-hESC migration into the scratched area.

(ectodermal) and cardiac (mesodermal) lineages. To investigate neural lineage specification, H9-hESCs and V-H9-hES cells were dissociated into small aggregates and then replated on laminin plates in neural stem cell (NSC) media. The expression of the NSC marker nestin was tested by immunofluorescence analysis. Nestin expression was ubiquitous in H9-hESC and V-H9-hESC neural precursors; however, the staining was more intense in variant-derived neural precursor cells (Fig 2A). To comparatively assess neural lineage multipotency, neural precursors derived from hESCs and V-H9-hESCs were differentiated towards neurons and astroglial lineage as depicted in schema (Fig S2A). An equal number of neural precursors were differentiated towards neurons after EGF and bFGF withdrawal. Tuj1 staining revealed no significant difference in the number of neurons (Fig 2B). For astrocytic differentiation, neural precursor cells were cultured in growth factor–depleted media and were exposed to BMP4. Glial fibrillary acidic protein (GFAP) is a reliable marker of astrocyte differentiation and maturation and has been extensively used to identify the astrocytic lineage (43). Therefore, we used GFAP to study the astroglial potential from normal versus variant human ES cells and noticed diminished GFAP+ astrocytes from V-H9-hESCs compared with H9-hESCs. Surprisingly, there were live cells in this growth condition without GFAP expression (Fig 2C). Furthermore, examining GFAP transcription revealed its expression was induced in H9-hESC–derived glial cells and was highly reduced in V-H9-hESC–derived glia (Fig 2D). This result corroborated with the GFAP protein data in Fig 2C. Pluripotent stem cells and NSCs were used as controls, which showed negligible transcripts of GFAP (Fig 2D).

Next, we examined the differences in the expression of key regulators of NSCs and astroglial differentiation. The transcription factor PAX6 is expressed in NSCs derived from hESCs, neural stem/progenitor cells (NSCs), astroglial cells, and throughout the central nervous system. PAX6 is known to positively regulate neurogenesis via controlling gene regulatory networks (44), and its role as an intrinsic fate determinant of the neurogenic potential of glial cells is also reported (45). Consistent with the previous reports, we observed elevated levels of PAX6 mRNA in NSCs (derived from both H9 and V-H9). PAX6 expression was seen in H9-hESC–derived glia and was substantially higher in V-H9-hESC–derived glial cells (Fig 2E). Although our data were consistent for neurogenic potential of V-H9-hESCs, we still did not understand the hampered glial commitment.

To address this, we next studied transcription factor nuclear factor IA (*NFIA*), which acts as a molecular switch to trigger human glial differentiation (46, 47). The transient expression of *NFIA* is sufficient to induce glial differentiation of NSCs derived from human pluripotent stem cells within 5 d and to convert these cells into astrocytes (46, 47). Upon assessing the expression of NFIA, we identified that unlike H9-hESC–derived glia, its levels were highly diminished in V-H9-hESC–derived glial cells and were similar to those seen in pluripotent and NSCs (Fig 2F). Collectively, these results demonstrate that V-H9-hESCs display aberrant differentiation features, particularly towards the astrocytic lineage because of the misexpression of key transcription factors such as PAX6 and NFIA.

We then tested the potential of V-H9-hESCs to differentiate into mesodermal lineage using directed differentiation towards cardiac lineage. Towards this, we derived cardiomyocytes from control H9-hESCs and V-H9-hESCs using the embryoid body (EB)–mediated differentiation protocol. Control EBs started beating after 15 d of induction; however, EBs derived from V-H9-hESCs were cystic with defined edges and failed to generate beating cells (Fig 2G). To examine the differences in the expression profile of key regulators of cardiac differentiation, we collected the EBs and measured the expression of early mesoderm-specific genes and cardiac-specific genes. The control (beating) and V-H9-hESC–derived EBs expressed the mesoderm-specific genes such as *Brachyury*, *Tbx6*, and *Mesp1*. The control (beating) and variant-derived EBs expressed the mesoderm-specific genes Brachyury, Tbx6, and Mesp1. However, ISL1, which identifies cardiac progenitor population, was weakly induced in V-H9-hESC–derived EBs. Cardiac-specific genes actin, TnnT, and Myh, which are required for its normal structure and function, were completely absent in variant-derived EBs (Fig 2H). These results suggest that V-H9-hESCs can differentiate into mesodermal precursor cells but fail to give rise to cardiac cell types.

Teratocarcinomas are characterized by the presence of both somatic tissues and undifferentiated malignant embryonal carcinoma cells (48). Using the teratoma assay, we previously reported that the V-H9-hESC–derived teratomas exhibited differentiation into all three germ layers (10). Next, we tested the retention of pluripotency markers such as Oct4 and Sox2 in teratoma tissue by qRT–PCR and immunostaining analysis (Fig S2B and C). Teratomas generated by V-H9-hESCs contained regions of Oct4+ and Sox2 cells, demonstrating the presence of undifferentiated stem cells (Fig S2D and E). Taken together, V-H9-hESC attributes such as enhanced cell proliferation, EMT, dysregulated differentiation, and retention of Oct4 and Sox2 overlap with neoplastic progression phenotypes.

### V-H9-hESCs exhibit altered genetic and epigenetic landscape

Genetic and epigenetic alterations play a crucial role in development and disease progression (49). Using karyotype analysis, we had demonstrated no apparent chromosomal anomalies in V-H9-hESCs (10). Next, we decided to detect DNA sequence copy-number variations that are undetectable by the standard cytogenetic assay using array comparative genomic hybridization. Genomic DNA of V-H9-hESCs demonstrated several small amplifications and deletions on multiple genes on chromosomes including chromosomes 1, 3, 12, and 21 (Table S1). Genes that exhibited anomalies such as T*nnt2*, *ICAM3*, *ROBO2*, *SLITRK1*, *BCL2L2* are not associated with stemness but are rather critical for cell differentiation and/or proliferation (Fig S3A). We further validated the expression of genes (ROBO2 and SLITRK1) that were genetically amplified in V-H9-hESCs by quantitative PCR analysis

---

V-H9-hESC showed significantly increased migration speed compared with H9-hESCs (scale bar = 20 $\mu$m). **(H)** Wound closure was expressed as the remaining area uncovered by the cells. The scratch area at time point 0 h was set to 100% (n = 4–6).

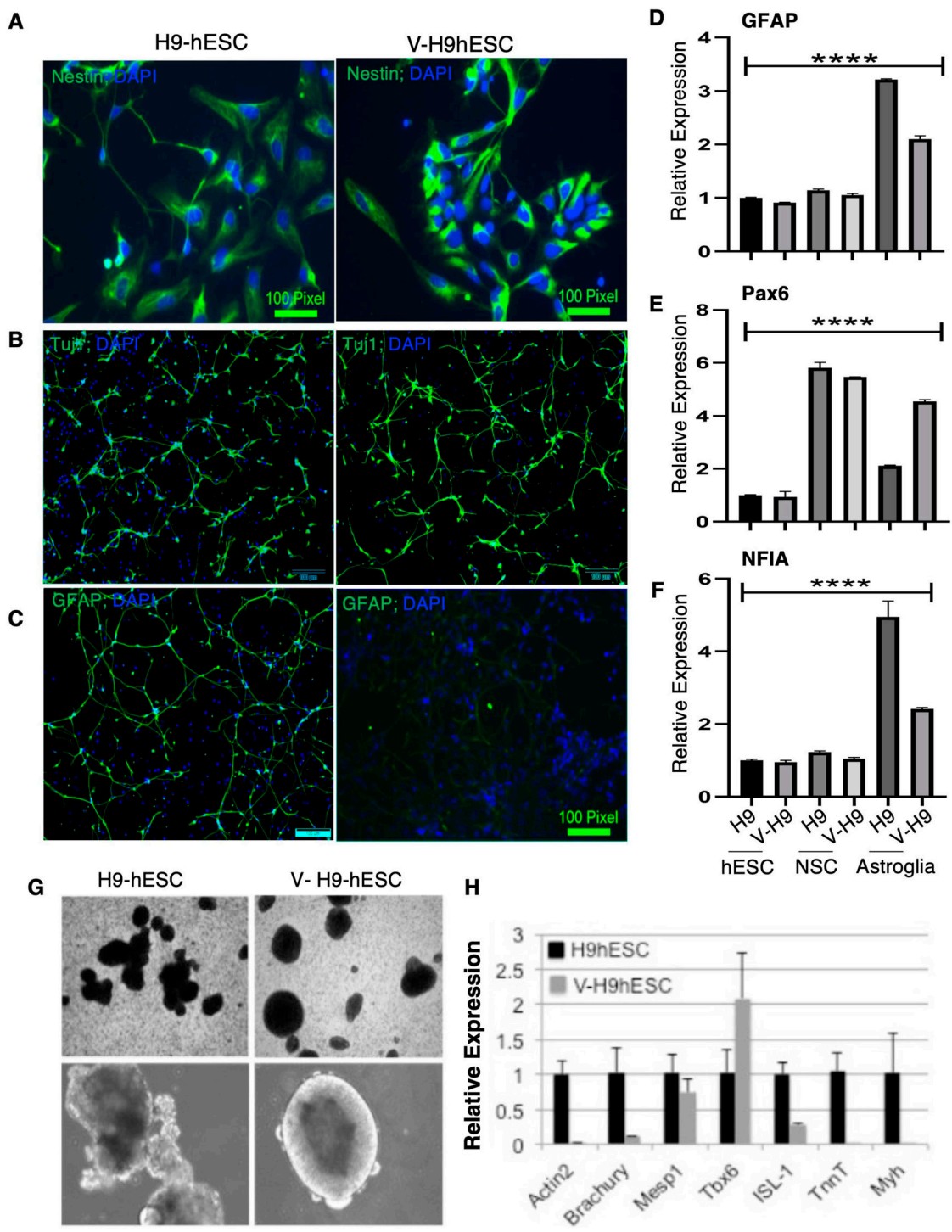

**Figure 2. Detailed characterization of the variant human ES line.**
**(A)** Immunostaining for nestin in H9-hESCs (scale bar = 100 pixels) and V-H9-hESCs (scale bar = 100 μm). **(B)** Immunostaining for Tuj1 in H9-hESCs and V-H9-hESCs (scale bar = 100 μm). **(C)** Immunostaining for GFAP in H9-hESCs and V-H9-hESCs (scale bar = 100 μm). **(D, E, F)** Quantitative RT–PCR for the indicated genes in pluripotent stem cells, neural stem cells, and differentiated astroglial cells (n = 3 for H9-hESCs, V-H9-hESCs, H9-NSCs, V-H9-NSCs, and H9- and V-H9-derived astroglial cells). Error bars: SEM. Ordinary one-way ANOVA, ****P < 0.0001. **(G)** Phase-contrast images of EBs during directed cardiac differentiation of H9-hESCs and V-H9-hESCs. **(H)** Relative expression of mesodermal- and cardiac-specific genes.

(Fig S3B). Significantly enhanced abundance of ROBO2 and SLITRK1 confirmed amplification of these genes in V-H9-hESCs. SLITRK5 did not show any difference in gene expression indicating the mutation did not have any impact on the expression of this gene (Fig S3B).

A number of variant lines have been characterized worldwide and reported to have overlapping and distinct chromosomal alterations (50). However till date, no single genomic marker has been known to be commonly/consistently altered in all these lines. Genetic changes can enhance lineage plasticity of the cell phenotype to take on a different molecular identity; however, it is also controlled by transcriptional and epigenetic regulation (51). To investigate possible alteration in the epigenomic landscape of V-H9-hESCs, we mapped various histone modifications that are predominantly studied in the regulation of the E-cadherin gene along with the enzymes that deposit these marks. Towards this, we performed Western blot analysis for H3K27me2, H3K9me2, H3K36me2, and H3K27Ac modifications. Our results demonstrated elevated levels of H3K27me2, H3K9me2, and H3K36me2; on the contrary, H3K27Ac was diminished in V-H9-hESCs indicating the altered epigenomic landscape. Total histones and GAPDH remained unchanged in H9-hESCs and V-H9-hESCs (Fig 3A).

The euchromatic histone methyltransferase (EHMT1 and EHMT2) complex deposits H3K9me2 on histones. Elevated levels of EHMTs in V-H9-hESCs attested to increased H3K9me2 marks (Fig 3B and C). Ezh2, a catalytic subunit of the PRC2 complex, did not show any significant alterations in expression and did not correlate with enhanced H3K27me2 marks (Fig 3B).

Mono- and di-methylation of H3K36 deposited by NSD proteins (NSD1, NSD2, and NSD3) play an important role in maintenance of chromatin integrity and regulate the expression of genes that control cell division, apoptosis, DNA repair, and EMT (52, 53, 54). Mutations or the aberrant expression of the NSD family is associated with developmental defects and cancers. Particularly, NSD2 and NSD3 are frequently overexpressed in various cancers (55). Interestingly, all three NSD family members that were up-regulated were enriched in V-H9-hESCs (Fig 3B and C) compared with H9-hESCs. We noticed very low levels of NSD1 and NSD3 (both long and short forms) in H9-hESCs compared with variants. NSD2 (three isoforms of 140, 65, and 58 KD) (56) were induced in V-H9-hESCs and were undetectable in H9-hESCs (Figs 3B and S3C). Given NSD3 has gained considerable attention in cancer development and progression (57), we decided to focus our efforts in understanding the role of NSD3 in hESC transformation.

We started by studying the potential mRNA- and/or protein-level dysregulation of NSD3 and EHMTs. Quantitative PCR analysis in H9-hESCs and V-H9-hESCs demonstrated no significant changes in transcription of above-mentioned enzymes; however, there were small but significant changes in splicing of the NSD3 gene, which is evident from enhanced levels of the NSD3-short form (Fig 3C). These results indicated mRNA levels might not be the determinants of protein levels of the key KMTs related to V-H9-hESCs.

To investigate the additional changes that are responsible for V-H9-hESC state, we profiled gene expression in H9-hESC and V-H9-hESCs from three different replicates. Unsupervised hierarchical clustering using principal component analysis identified two distinct groups of normal and V-H9-hESCs (Fig S3D). Comparative RNA-Seq analysis identified that ~8% genes were differentially regulated among the two lines tested. Out of 7,482 genes, 284 (4.6%) genes were up-regulated, and

345 (3.8%) genes were down-regulated (false discovery rate < 0.05) (Fig S3E and F). The volcano plot illustrates the statistically significant fold change of differentially expressed genes in the normal versus variant pluripotent stem cells (Fig 3D and Table S2). Specifically, down-regulation of E-cadherin and up-regulation of snail attest to the previous observation of the EMT phenotype in V-H9-hESCs (Fig 3D). Gene ontology analysis identified that processes such as migration, neuronal differentiation, chromatin organization, RNA Pol 2–mediated transcription, and protein ubiquitination are altered in H9- versus V-H9-hESCs, which is consistent with various functional data such as EMT, stability of chromatin modifiers, and aberrant astroglial differentiation (Figs 3E and S3G). We also noticed up-regulation of the BCL2L2 transcript, although the gene contains deletion mutation (Fig 3E and Table S2), indicating that the mutation data did not correlate with gene expression for BCL2L2. Nonetheless, identification of discrete chromosomal alteration coupled with epigenetic and cellular changes in V-H9-hESCs testifies these cells have undergone transformation.

## EHMTs interact and methylate NSD3 protein in variant hESCs

Protein lysine methyltransferase and modifications deposited by them on histones form a highly interconnected network to bring about changes in the epigenetic landscape. Therefore, disturbing one component may rearrange the entire system. For example, it is well known that NSD3 catalyses the methylation H3K36 by binding to the complex of the LSD2/EHMT2 protein (58). Given that V-H9-hESCs showed the overexpression of EHMTs and NSD3, we decided to explore whether there is a relationship between the two chromatin modifiers in variant hESCs. We began by investigating whether EHMTs and NSD3 crosstalk to each other by physically interacting with each other. Towards this, we performed immunoprecipitation with anti-EHMT1/anti-EHMT2 antibodies in V-H9-hESC lysates and immuno-blotted with anti-NSD3 antibodies (Fig 4A). Appearance of NSD3 isoforms (NSD3L and NSD3S) confirmed the association between chromatin modifiers in variants (Fig 4A and B). Compared with pluripotent cells, HEK293 cells are amenable to transfection, which would allow us to explore the molecular relationship between the two lysine methyltransferases by overexpression or knockdown studies. Therefore, we tested whether these two proteins associate with each other in heterologous 293 cell system by overexpressing V5-EHMT1 and NSDL-GFP/V5-EHMT1 and Flag-NSDS. Lysates prepared from the transfected cells were incubated with anti-V5 antibody or anti-Flag antibody, and the immunoprecipitates were subjected to Western blot analysis. The use of anti-GFP and anti-Flag antibodies clearly showed that EHMT1 associates with NSD3 isoforms (Fig S4A and B). We also performed co-immunoprecipitation with anti-EHMT1 and anti-EHMT2 antibodies and immunoblotted with anti-NSD3 antibodies in 293s to confirm endogenous protein interactions. Both overexpression and endogenous IPs revealed EHMT1 interacted with NSD3 (Fig S4C and D). Overall, EHMT1/2 IP and reciprocal IP with anti-NSD3 antibodies confirmed the association of EHMTs with NSD3 in both variant and 293 cells. Furthermore, immunoprecipitation of endogenous EHMT1 (IP) and excision of unique bands, which were subjected to LC/MS analysis, identified LMNB1 (59), histones, and NSD3 as interactors of EHMT1 (Fig S4E).

Because NSD3 is interacting with euchromatic histone methyltransferases, we tested whether NSD3 is a substrate for methylation;

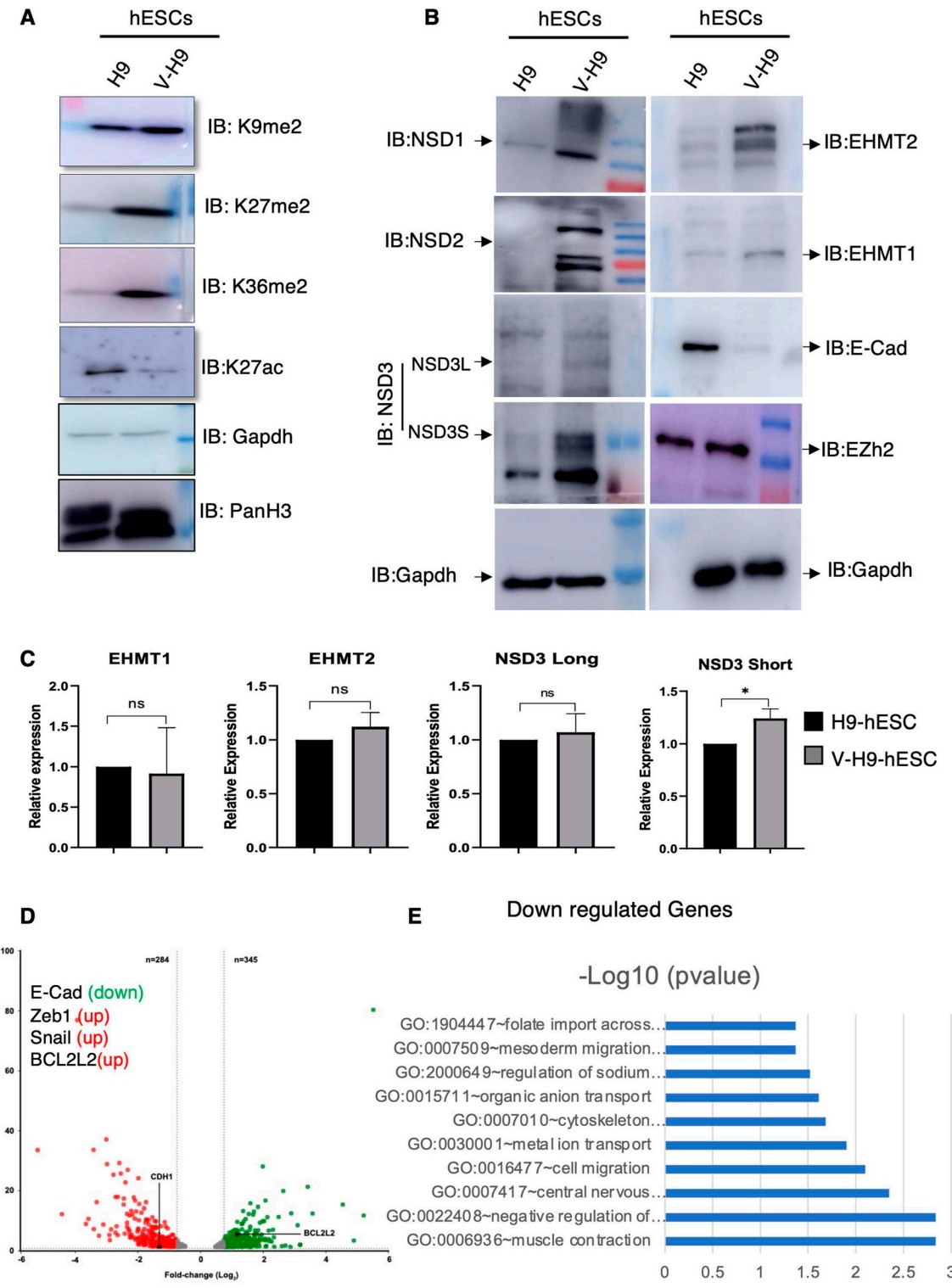

**Figure 3. Characterization of epigenomic patterns in the variant human ES line.**
**(A)** Western blotting analysis of H3K9me2, H3K27me2, H3K27ac, and H3K36me2 in H9-hESCs and V-H9-hESCs. **(B)** Western blotting analysis of E-cadherin, EHMT1/2, Ezh2, NSD1, NSD2, NSD3L, and NSD3S in H9-hESCs and V-H9-hESCs. **(C)** qRT-PCR analysis of EHMTs and NSD3 transcripts in H9-hESCs and V-H9-hESCs. **(D)** Volcano plot demonstrating the significantly changed transcripts in V-H9-hESCs in comparison with H9-hESCs. **(E)** Gene ontology analysis of differentially regulated genes in H9-ESCs and V-H9-ESCs (down-regulated genes).

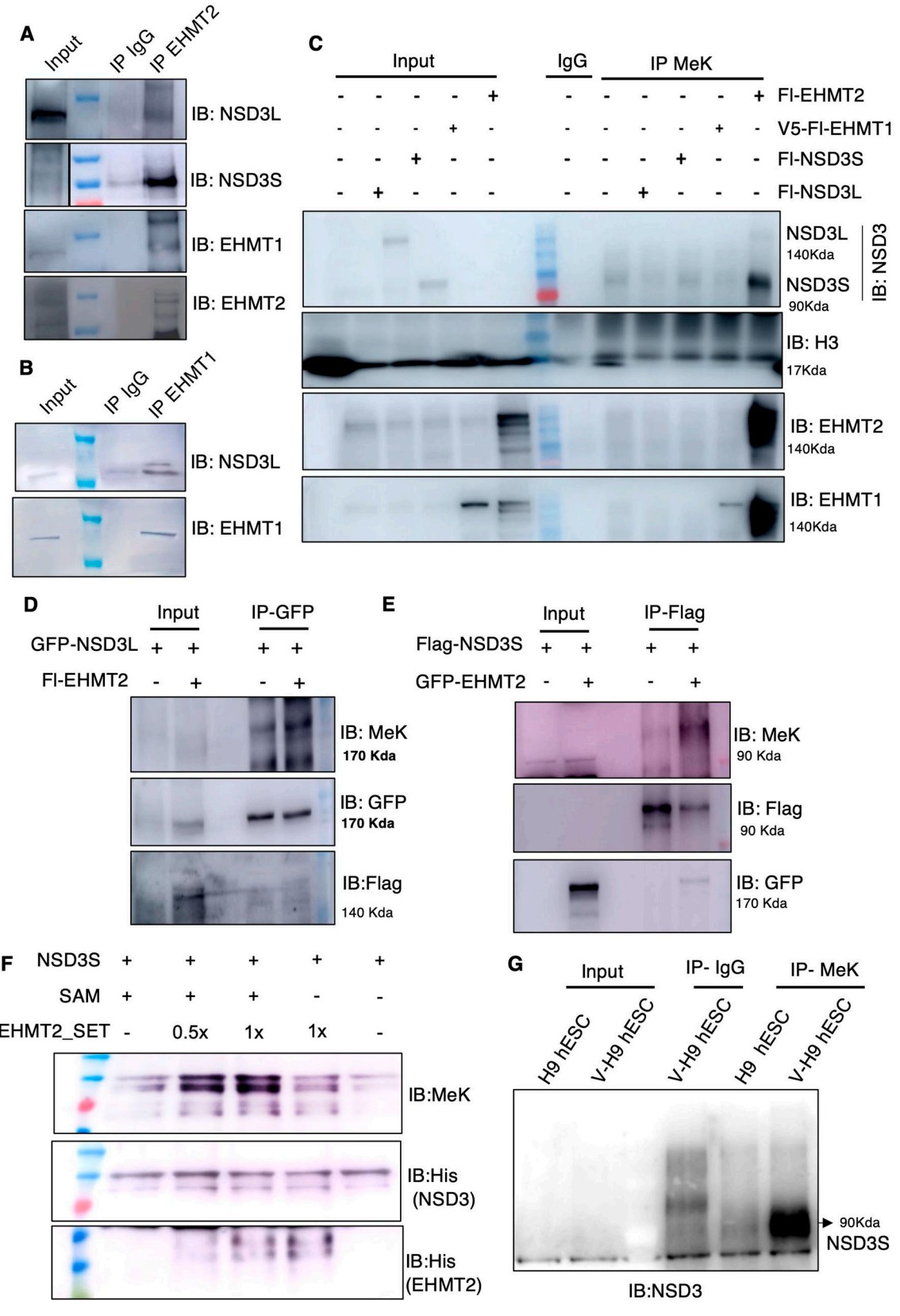

that is, lysine residues of NSD3 are methylated in cells. To examine the methylation of NSD3 in cells by EHMT1 and EHMT2 independently, we overexpressed EHMTs independently in the 293s and pulled down with anti-pan methyl-lysine (MeK) antibody and probed for NSD3 in the immunoblot. EHMT2-overexpressing cells (but not EHMT1) show more methylation of both NSD3-long and NSD3-short isoforms (Fig 4C). The overexpression of EHMT1 and EHMT2 was confirmed by immunoblotting with specific antibodies. PanH3 served as a positive control for MeK IP (Fig 4C). Taken together, both EHMTs interact with NSD3 in cells, but only EHMT2 methylates NSD3-long and NSD3-short isoforms.

To further confirm that EHMT2 indeed is responsible for NSD3 methylation, we co-transfected EGFP-NSD3L along with Flag-EHMT2 or Flag-NSD3S with EGFP-EHMT2 (Supplemental Data 1). Immunoprecipitations were performed by pulling the NSD3 protein using anti-GFP or anti-Flag antibodies. The amount of NSD3L and NSD3S immunoprecipitated was confirmed using anti-GFP antibody and anti-Flag antibody (Fig 4D and E). The overexpression of EHMT2 was confirmed by the indicated antibody. In these conditions, we observed that the overexpression of EHMT2 enhanced methylation of NSD3 isoforms; particularly, NSD3S was much strongly methylated.

We then performed the in vitro methyltransferase assay using recombinant NSD3S and EHMT2 SET (Su(var)3-9, enhancer of zeste, and Trithorax) domain proteins (Fig S4F–H). Towards this, we used different concentrations of EHMT2 SET and incubated with NSD3S in the presence or absence of co-factor SAM (S-adenosyl methionine). When products of these reactions were immunoblotted using the methyl-K antibody, the specific methylation signal on NSD3S increased with increasing amounts of EHMT1/2-SET in the presence of SAM (Fig 4F). This result further confirmed NSD3 as a substrate for methylation by EHMT2 in vitro and in vivo. Considering our finding that NSD3 is induced and stabilized at protein levels in V-H9-hESCs, we speculated that NSD3 is methylated in V-H9-hESCs. Immunoprecipitation of NSD3 in H9-hESCs and V-H9-hESCs followed by probing with anti-MeK antibody indicated NSD3; specifically, NSD3S was heavily methylated in variants and its levels were very high in variants (Fig 4G). Taken together, our data demonstrate that EHMT2 stabilizes NSD3 protein in V-H9-hESCs via methylation.

## NSD3 stabilization mediates transformation in human pluripotent stem cells

The epigenetic landscape regulated by NSD3, EHMT1, and EHMT2 controls diverse biological processes, and their deregulation is associated with various diseases (52, 60, 61). These enzymes are also associated with processes such as cellular proliferation,

metabolic adaptation, EMT, metastasis, and stem cell maintenance that promote cancers (62, 63). In this study, we observed elevated levels of H3K9me2 and H3K36me2 marks along with an increase in the respective lysine methyltransferase (EHMT1, EHMT2, and NSD3) in V-H9-hESCs. Among these, methyltransferase NSD3 encodes two splicing variants, NSD3-short (NSD3S) and NSD3-long (NSD3L) isoforms, which are frequently amplified in cancers (55). NSD3L exerts its effect via mono- and di-methylation of H3K36; NSD3S containing PWWP domain and acidic transactivation domain influences biological functions via protein–protein interactions (55, 64, 65). To test the influence of NSD3 methylation and stability in transformation of hESCs to the variant state, we decided to knock down the NSD3 in V-H9-hESCs.

Towards this, we knocked out the *NSD3* gene in V-H9-ESCs by the dual-guide RNA-mediated CRISPR-Cas9 strategy (Fig S5A). To validate the *NSD3* knockdown, the plasmid with an EGFP reporter targeting the *NSD3* gene was transfected in 293 cells and GFP-positive cells were sorted and further amplified. PCR genotyping was done in which a 328-bp-long product confirmed the deletion of exon 2 from the *NSD3* gene compared with a 3.1-Kb-long product in control (Fig S5B).

We started by testing the impact of depletion of NSD3 on molecular and phenotypic changes in V-H9-hESCs by CRISPR-mediated genetic depletion of NSD3 using the dual-guide RNA strategy (Fig S5A). NSD3 gRNA-GFP reporter–containing plasmids were transfected in V-H9-hESCs, and GFP-positive cells were sorted and cultured to obtain a V-H9-hESC<sup>NSD3-KD</sup> line (Fig S5C). The genotype assay was performed using a specific primer, which will amplify the fragment of 3.1 Kb size (in unedited condition) and 328 bp upon deletion of exon 2 of NSD3-long and NSD3-short isoform proteins confirming gene editing (Fig S5C). Consequent to the NSD3 depletion (long and short isoforms), we noticed loss of H3K36me2 in V-H9-hESCs<sup>NSD3-KD</sup> (Fig S5D and E). Morphological analysis of V-H9-hESC<sup>NSD3-KD</sup> colonies demonstrated tightly packed round cells without spaces between the cells (Fig 5C). These colonies were similar to normal H9-hESCs and were, unlike V-H9-hESCs colonies, comprised of flattened mesenchymal-like cells (Fig 5A–C). This observation indicated the EMT phenotype noticed in variant hESCs must have reversed towards the epithelial state. To confirm this, we performed immunostaining for epithelial marker E-cadherin and EMT marker vimentin in H9-hESCs, V-H9-hESCs, and V-H9-hESCs<sup>NSD3-KD</sup> (Fig 5D). E-cadherin expression was highly diminished in V-H9-hESCs but was regained in cell-to-cell contacts of V-H9-hESCs<sup>NSD3-KD</sup> similar to H9-hESCs (Fig 5D). We also noticed the diminished expression of vimentin in V-H9-hESCs<sup>NSD3-KD</sup> compared with V-H9-hESCs. Interestingly, levels of vimentin were lower than

**Figure 4. EHMT2 interacts and methylates NSD3 in variant human ES cells.**
**(A)** Immunoprecipitation of EHMT2 antibody followed by Western blotting for indicated protein in V-H9-hESCs. The black dotted line represents input of the different exposure of the same gel. **(B)** Immunoprecipitation of EHMT1 antibody followed by Western blotting for indicated protein in V-H9-hESCs. **(C)** 293 cells were transfected with indicated plasmids, immunoprecipitated with methylated lysine antibody, and immunoblotted for indicated antibodies. **(D)** 293s were transfected with indicated plasmids, immunoprecipitated with GFP antibody, and immunoblotted for pan methylated lysine antibody, and the same blot was reprobed for total protein using GFP antibody. **(E)** 293s were transfected with indicated plasmids, immunoprecipitated with Flag antibody, and immunoblotted for pan methylated lysine antibody, and the same blot was reprobed for total protein using Flag antibody. **(F)** Purified 8xHis-NSD3 was incubated overnight at 37°C along with SAM and an increasing concentration of purified EHMT2 SET domain. Reaction mixture was immunoblotted for the indicated antibodies. **(G)** Immunoprecipitation with anti-methylated lysine-specific antibodies and immunoblotting for NSD3 in H9-hESCs and V-H9-hESCs.

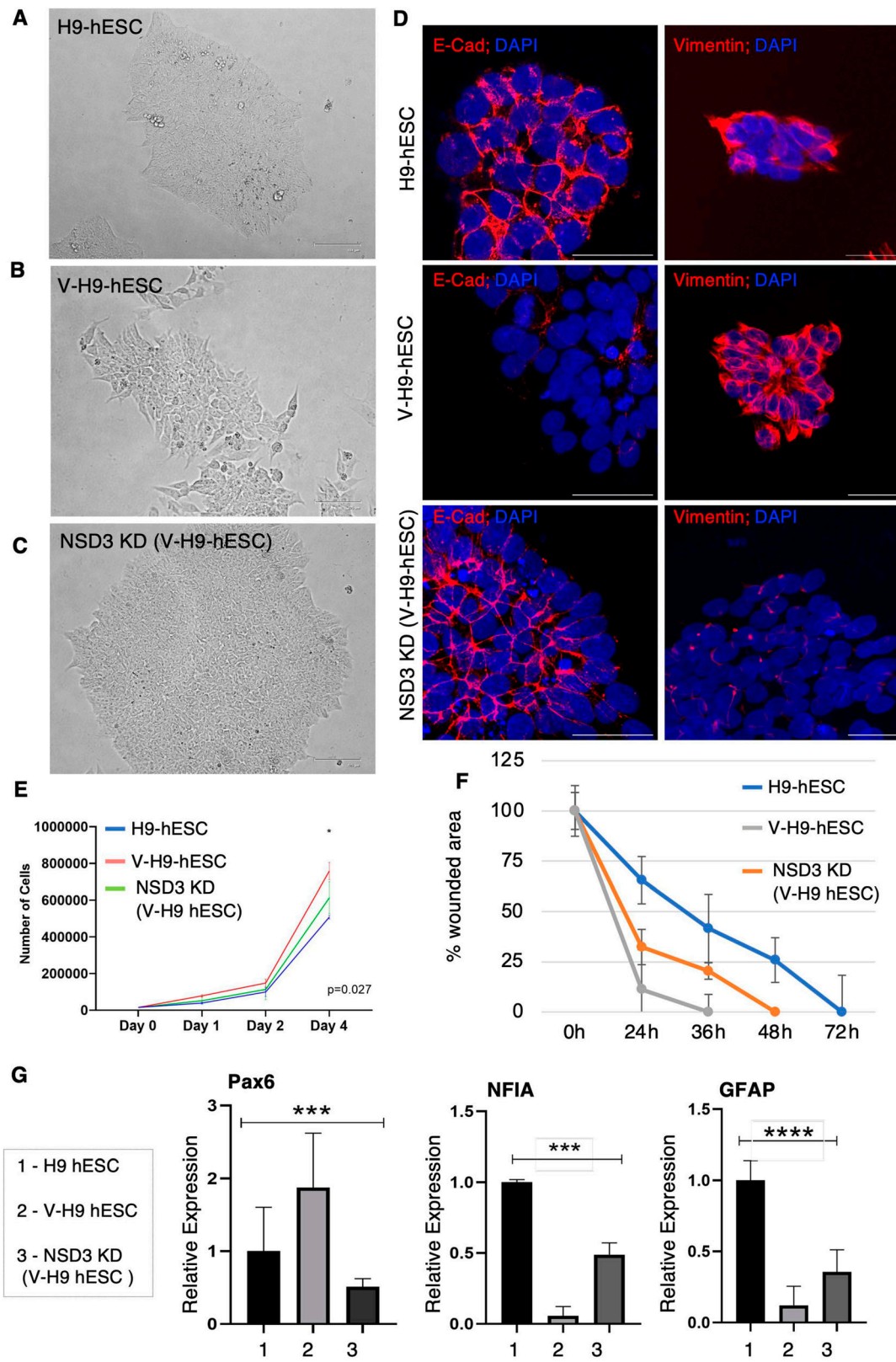

**Figure 5. Depletion of NSD3 reverts morphological phenotypic defects seen in variant human ES cells.**
**(A, B, C)** Phase-contrast image depicting the morphology of H9-hESCs, V-H9-hESCs, and V-H9-hESCs^NSD3-KD (scale bar = 125 $\mu$m). **(D)** Immunostaining of E-cadherin and vimentin in H9-hESCs, V-H9-hESCs, and V-H9-hESCs^NSD3-KD (scale bar = 40 $\mu$m). **(E)** Rate of proliferation of H9-hESCs, V-H9-hESCs, and V-H9-hESCs^NSD3-KD was studied in a cell proliferation assay. **(F)** Wound closure expressed as the remaining area uncovered by the cells. The scratch area at time point 0 h was set to 100% (n = 4–6).

normal H9-hESCs (Fig 5D). Acquisition of E-cadherin in V-H9-hESCs<sup>NSD3-KD</sup> was also confirmed by Western blot analysis (Fig S5D).

Given the key feature of the V-H9-hESCs is faster growth over H9-hESCs, we next examined the proliferation status of V-H9-hESCs<sup>NSD3-KD</sup>. To this end, we seeded an equal number of H9-, V-H9-hESCs, and V-H9-hESCs<sup>NSD3-KD</sup> and estimated the cell count over 4 d. As shown previously, the growth rate profiles of V-H9-hESCs were accelerated compared with the parent line. Conversely, V-H9-hESCs<sup>NSD3-KD</sup> grew significantly slower than V-H9-hESCs but were comparatively faster than H9-hESCs (Fig 5E). In addition, we also noticed that reduction in NSD3 levels in V-H9-hESCs<sup>NSD3-KD</sup> was accompanied by reduction in c-Myc levels (Fig S5F). This is consistent with previous reports where NSD3S is shown to bind and stabilize the c-Myc protein (66). Investigation into the levels of NSD1 and NSD2 in V-H9-hESCs<sup>NSD3-KD</sup> revealed that the NSD1 protein was reduced in these cells; however, NSD2 levels remained unchanged (Fig S5G).

Next, we tested migration of V-H9-hESCs<sup>NSD3-KD</sup> compared with H9-hESCs and V-H9-hESCs in the scratch assay. As expected, V-H9-hESCs migrated and rapidly proliferated in the scratched area in 24 h. The scratched area was then confluent by 36 h. However, V-H9-hESCs <sup>NSD3-KD</sup> closed the wounds at 48 h, which were albeit slower than V-H9-hESCs and were faster than H9-hESCs (Fig 5F). Interestingly, the pattern of V-H9-hESC<sup>NSD3-KD</sup> migration resembled H9-hESCs; that is, the cell at the edge migrated in the form of a sheet. Proliferation and migration data indicated that these properties were partially reverted upon NSD3 knockdown in V-H9-hESCs.

The contribution of elevated NSD3 in compromised astroglial differentiation in V-H9-hESCs was studied by differentiating the V-H9-hESC<sup>NSD3-KD</sup> line towards astroglial lineage. Towards this, H9-hESCs, V-H9-hESCs, and V-H9-hESCs<sup>NSD3-KD</sup> were differentiated into glial cells as described in Fig S2A and qRT–PCR analysis was performed for transcription regulators PAX6, NFIA, and GFAP. Our results identified that in V-H9-hESCs<sup>NSD3-KD</sup>, the aberrant expression of these genes was largely rescued (Fig 5G) confirming the role of NSD3 in astroglial differentiation (Fig 5G). Overall, these results indicate that depletion of NSD3 reverts the morphological and molecular defects noticed in V-H9-hESCs and was responsible for transformation of H9-hESCs towards the variant state. Based on these, we conclude that NSD3 is a major contributor in transformation of H9-hESCs.

## PWWP domain of NSD3 is essential for interaction with EHMT2

Because EHMT2 showed association with both NSD3 isoforms, we hypothesized domains present in the short isoform and sufficient for interaction. At the structural level, NSD3S contains a small transactivation domain and PWWP domain, which binds to both histone and DNA in the nucleosome template regulating diverse biological functions (67). Towards this, we cloned different lengths of the NSD3 protein with epitope tags as indicated in the schema (Fig S6A). Tagged plasmids, Flag-EHMT2, were co-transfected with

NSD3L-GFP, and EHMT2-GFP was co-transfected with Flag-NSD3S constructs. Immunoprecipitation of EHMT2 using Flag antibody or GFP in co-transfected cells showed the association with both NSD3L and NSD3S (Fig 6A and B). Next, the domain-deleted NSD3 constructs were co-transfected with Flag-tagged EHMT2 in 293 cells. Anti-Flag IP was performed revealing a strong interaction of EHMT2 with N-term NSD3 (1–384) (Fig 6C). We observed no interaction of EHMT2 with the NSD3 N-term (1–173) domain (Fig 6D). The PWWP domain of NSD3 is known to bind histone H3K36 methylation; also, it contains four RK motifs, which are preferred motifs for methylation by EHMTs. Therefore, we designed primers to delete the 185–401 region of NSD3 (Fig S6B) and questioned (a) whether the PWWP domain is necessary for the interaction of NSD3 with EHMT2 and (b) whether EHMT2 methylated lysine/s in the PWWP domain. First, we performed an IP assay to check for the interaction between PWWP-deleted NSD3-long isoform fused with EGFP and wild-type NSD3L isoform fused with EGFP. When NSD3L wild-type and NSD3L PWWP mutants were co-transfected with Flag-tagged EHMT2 and pulled down with anti-GFP antibody, we observed a strong interaction of Flag-tagged EHMT2 with NSD3L isoform, but PWWP mutant weakly interacted with EHMT2 (Fig 6E) indicating the importance of the PWWP domain for associating with EHMT2, whereas methylation levels of wild-type versus PWWP mutant NSD3-long isoform remained comparable to those of the input amount. These findings indicate that lysine moiety that is targeted for methylation lies outside the PWWP domain.

## EHMT2-mediated methylation on K477 stabilizes the NSD3 protein

Next, we aimed to identify the lysine residue/s that is/are methylated in the NSD3 protein. Because both the isoforms were getting methylated in response to EHMT2 overexpression, we speculated that amino acids undergoing methylation must be shared between NSD3L and NSD3S. Upon performing in silico methylation site prediction as described by reference 68, we identified K477 lysine of NSD3 has higher propensity to be methylated based on the accessible surface area surrounded by modification sites (Fig S7A). This is consistent with our domain deletion and methylation analysis where we observed no change in methylation upon deletion of N-terminus and PWWP domains (Fig 6E). Interestingly, we observed the presence of K477 was flanked by AR and S motifs similar to H3K9-like ARKS motifs in both long and short isoforms of NSD3 and it is also evolutionarily conserved among lower to higher eukaryotic organisms (Fig S7B). Next, we decided to perform in vitro methylation assay where the enzyme and substrate are present surplus in the tube, and devoid of other complexities/competitions, the reaction proceeds irrespective of these binding domains. Towards this in vitro methyltransferase assay using Enzo Life Sciences, the HMT assay kit on the NSD3 peptide (VKIAWKTAAARKSLPASITM) containing K477 at the centre along with the lysine- to alanine-substituted mutant (K477A—VKIAWKTAAARASLPASITM) of the NSD3 peptide was employed. We observed that EHMT1/2 SET domains

---

**(G)** Quantitative RT–PCR for the indicated genes in differentiated astroglial cells derived from H9-hESC, V-H9-hESC, and V-H9-hESC<sup>NSD3-KD</sup> lines (n = 3). Error bars: SEM. Ordinary one-way ANOVA, PAX6, ***$P$=0.0003; NF1a, ***$P$=0.0001; And GFAP, ****$P$ < 0.0001.

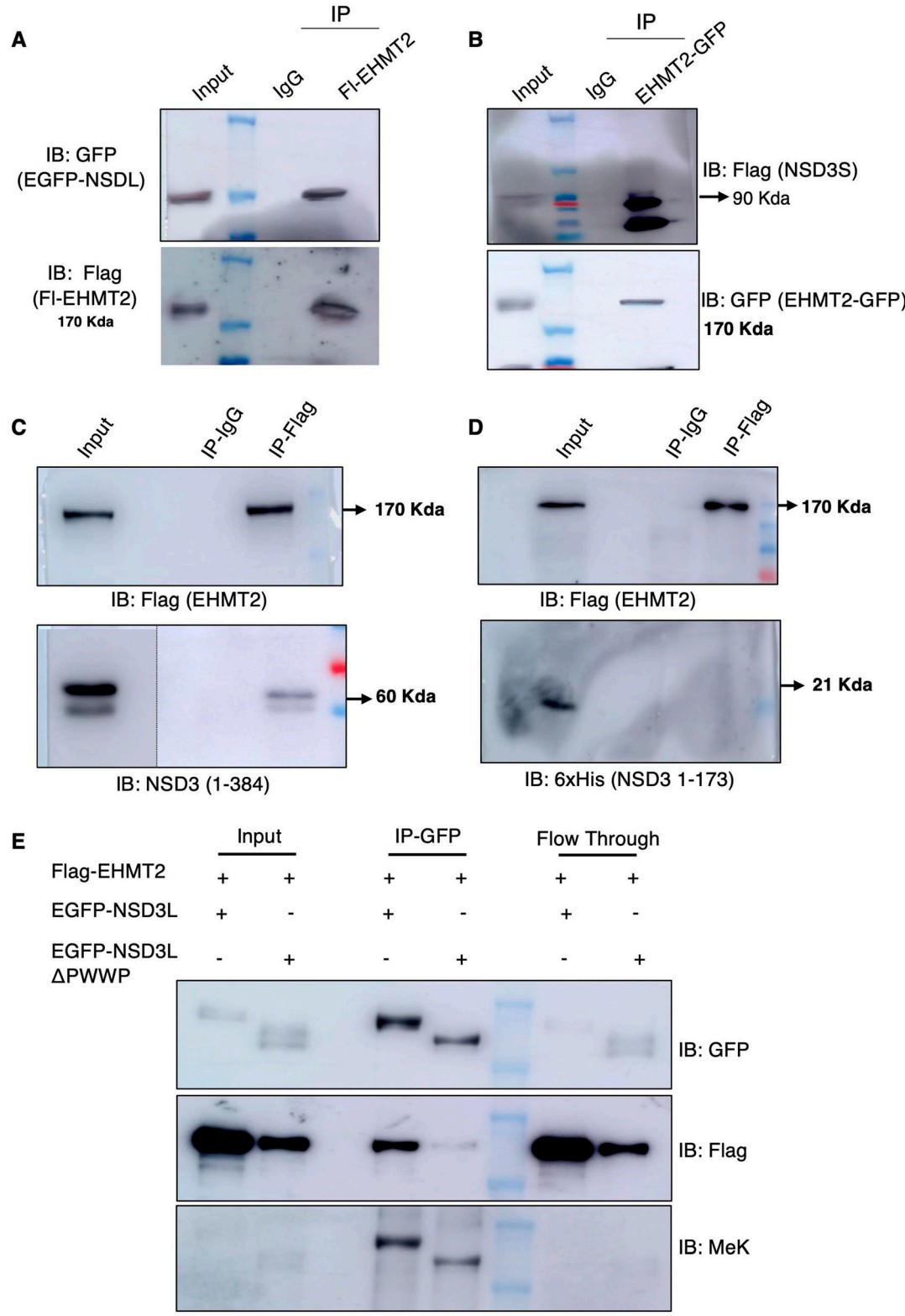

**Figure 6. PWWP domain of NSD3 is necessary for interaction with EHMT2.**
**(A)** Immunoprecipitation of Flag EHMT2 followed by immunoblotting with GFP antibody shows NSD3L-GFP with Flag-EHMT2. **(B)** Immunoprecipitation of GFP followed by immunoblotting with Flag antibody shows Flag-NSD3S with EHMT2-GFP. **(C)** Immunoprecipitation of Flag-EHMT2 followed by immunoblotting with NSD3 antibody shows NSD3 1-384 domain interacting with Flag-EHMT2. The black dotted line represents input of the different exposure of the same gel. **(D)** Co-immunoprecipitation of Flag-EHMT2 followed by immunoblotting shows no interaction of NSD3 1–173 domain with EHMT2. **(E)** Immunoprecipitation of EGFP-NSD3L or EGFP-NSD3L delta PWWP using

could methylate K477 of the NSD3 peptide, whereas no methylation was detected on the K477A mutant peptide (Fig 7A and B). Also, upon treatment of BIX01294 as a substrate-competitive inhibitor of EHMT1/2, methylation was inhibited from the wild-type peptide (Fig 7A and B). Taken together, these data suggest that EHMT1 and EHMT2 interact with the NSD3 and methylate it at lysine K477.

Because lysine can be methylated at different degrees, viz., mono-, di-, and tri-methylation (69), it is crucial to identify the status of methylation on target lysine as they may have diverse functional consequences in the cells. To determine the status of lysine methylation on the NSD3 peptide, we used high-resolution NMR spectroscopy. In contrast to other methods that rely on indirect measurements of methylation and predefined experimental set-up, high-resolution NMR spectroscopy uses unique proton/carbon ($^{1}$H–$^{13}$C) correlation signals of mono-, di-, and tri-methylated lysine to detect the status of lysine methylation in an unbiased straightforward manner (70). Furthermore, precise detection of respective forms of lysine methylation during the course of methyltransferase reaction in a non-disruptive mode provides mechanistic insights into the activity of EHMT1/2. Therefore, we performed in vitro methyltransferase reaction with the NSD3 K477 peptide as the substrate with SAM and EHMT1/2 SET domain independently. At the beginning of reaction ($0^{th}$ hr), we did not detect any methylation peaks at 2D natural abundance $^{1}$H–$^{13}$C correlation spectra (Fig S7C and E). During the course of the reaction, we observed abundant mono-methylated lysine CH3ξ-Me1 resonances at ~3.1/51.0 ppm along with lysine methyl resonances at ~2.7/35.5 ppm (KMe1) and less abundant di-methylated lysine CH3ξ-Me2 resonances at ~3.2/60.0 ppm along with lysine methyl resonances at ~2.9/45.5 ppm (KMe2) (Figs 7C and S7F). Upon longer incubation, we observed the conversion of mono-methylated lysine into di-methylated lysine. Besides, there was a gradual build-up of tri-methylated lysine resonances at ~3.1/55.5 ppm (KMe2) after 12 h (Figs 7D and S7G). To confirm the position of lysine that gets methylated in the NSD3 peptide, we performed a control experiment with K477A-substituted alanine mutant. We did not observe any methylation peaks even after 16 h of incubation (Fig S7H). Together, these results indicate that EHMT1/2 SET preferentially mono-methylates NSD3 at K477 in vitro and upon longer incubation can perform di/tri-methylation. Unlike data in Fig 4C, peptide methyltransferase assay and NMR demonstrated that both EHMT1 and EHMT2 can methylate NSD3 in vitro.

Next, to check for the status of methylation of NSD3 in cells, we performed immunoprecipitation with anti-pan mono/di-methyl-lysine antibody using 293 cellular lysates and immunoblotted with anti-NSD3 antibody and anti-histone H3 antibody for positive control (Fig 7E). These results showed that both NSD3-long and NSD3-short isoforms are mono/di-methylated in vivo. We performed immunoprecipitation with anti-NSD3 antibody and immunoblotted with mono/di-methyl-lysine–specific antibodies (Fig 7F and G). These results showed that both NSD3-long and NSD3-short isoforms are mono/di/tri-methylated in vivo.

To gain insights into the role of K477 residue methylation on NSD3, we generated K477R mutants via site-directed mutagenesis using Flag-NSD3S cloned in a construct with IRES GFP. Lysates from 293 cells overexpressing the Flag-NSD3S or Flag-NSD3S K477R mutant were subjected to Western blot analysis and were probed for anti-GFP and anti-Flag antibodies. We found diminished levels of mutant NSD3 protein indicative of mutant protein being targeted for degradation (Fig 7H). Contrarily, the levels of GFP and GAPDH proteins remained unchanged. Overall, our data identified EHMT2-mediated methylation on K477 and its importance in stabilization of NSD3 isoforms.

## Protein degradation mechanism regulates NSD3 levels in normal pluripotent stem cells

Given EHMTs and NSD3 are associated with each other and NSD3 protein gets methylated by EHMT2 in cells, we investigated a possible relationship between the two methyltransferases in detail. We began by testing the effect of the overexpression of either EHMTs or NSD3 enzymes on the other partner. Towards this, we transfected EHMT1, EHMT2, and NSD3 transgenes individually in 293 cells and probed for the indicated proteins. Interestingly, we observed induction of NSD3 upon EHMT2 overexpression (Figs 8A and S8A); however, EHMT1 overexpression did not alter NSD3 levels (Fig S8B). In a vice-versa experiment, the overexpression of NSD3 did not influence EHMT2 levels (Fig S8C). The change in NSD3 protein levels upon EHMT2 overexpression was independent of transcription (Fig S8D). Further to confirm the relationship between EHMT2 and NSD3 proteins, EHMT1 and EHMT2 were depleted using specific shRNA and NSD3 protein levels were investigated. 293 cells stably selected for EHMT1 and EHMT2-specific shRNA demonstrated loss of EHMT2 protein compared with control shRNA-expressing cells (Fig 8B). We observed the loss of EHMT1 or EHMT2 protein was coupled with loss of NSD3 protein in these cells (Fig 8B). To test the involvement of EHMT2's catalytic activity on NSD3 protein levels, we treated 293 cells with known inhibitors of EHMT activity (UNC0638 and BIX01294). Immunoblot shows reduced NSD3L upon increased inhibition of EHMT1 catalytic activity confirming shRNA depletion data (Fig 8C). Interestingly, EHMT inhibitors did not alter the transcript levels (Fig S8E). Overall, our results indicated levels of NSD3 protein are directly correlated with higher levels of EHMT2 and its catalytic activity. Nonetheless, this correlation seemed to be independent of transcription.

Next, we speculated NSD3 may be a short-lived protein and subjected to degradation via the ubiquitin–proteasome system in cells. We also presumed that EHMT2-mediated NSD3 methylation may antagonize degradation of NSD3. To address this, 293s were preincubated with the 26S proteasome inhibitor MG132 or the translation elongation inhibitor cycloheximide (CHX), which has been widely used to measure degradation kinetics for short-lived proteins. To evaluate the half-life of NSD3, 293s were treated with 20 µg/ml of cycloheximide for different amounts of time indicated and cell lysates were analysed for amount of NSD3. Interestingly,

---

GFP antibody followed by immunoblotting shows the interaction of EHMT2 with full-length NSD3_Long isoform and less interaction with PWWP-deleted NSD3_Long_Isoform. Anti-MeK antibody probing showed no difference in methylation levels.

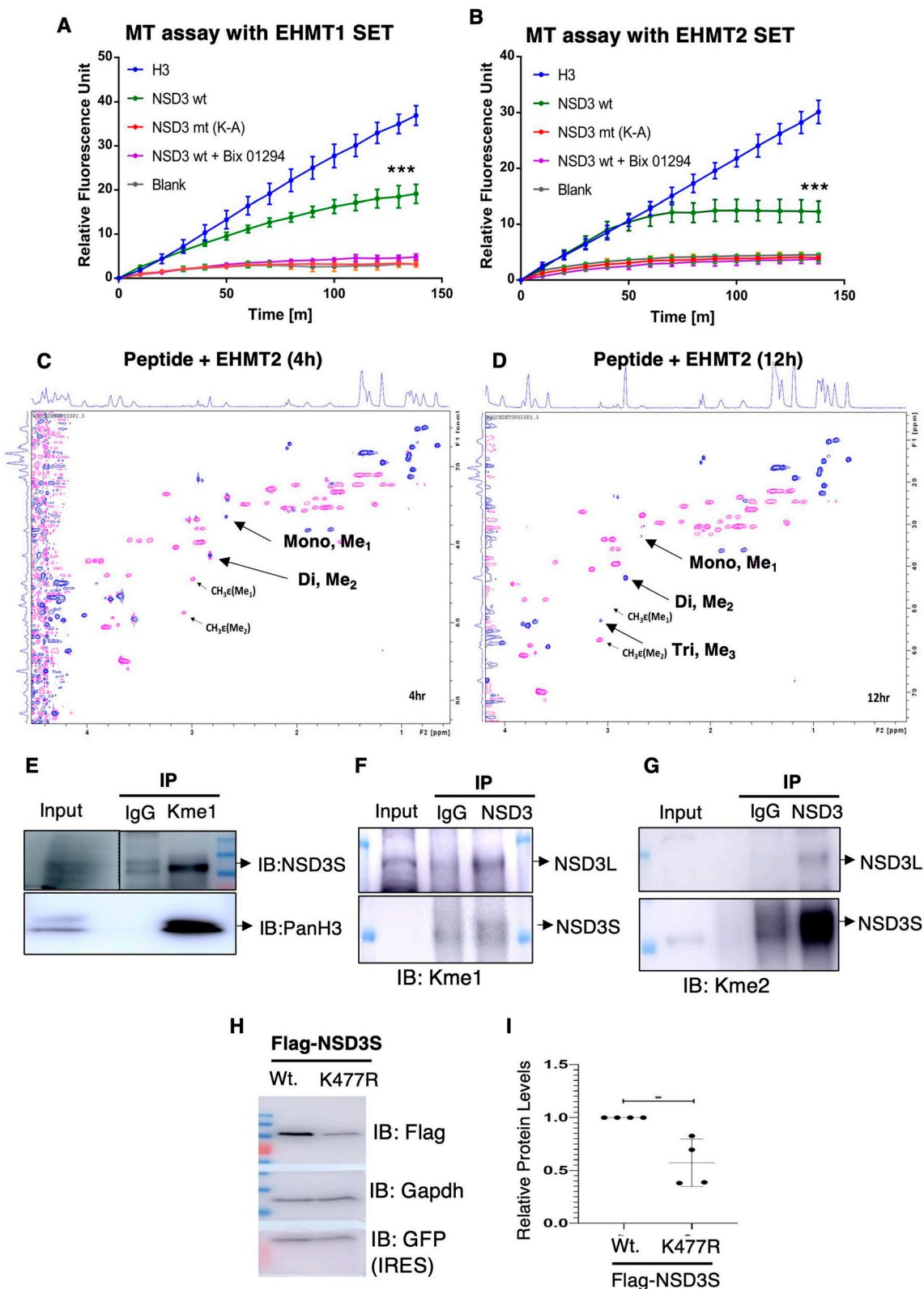

**Figure 7. EHMTs methylate NSD3 protein at K477 residue and stabilize its levels.**
**(A)** In vitro methylation assay performed using NSD3 wt & mt peptide and EHMT1 SET domain in the presence or absence of BIX01294 (n = 2). H3 peptide was used as a positive control (n = 2). Technical replicates were plotted with mean values. There is a significant difference between wt, mt, and BIX, $P < 0.001$ (Dunn's multiple comparison test). The error bar represents ± SD. **(B)** In vitro methylation assay performed using NSD3 wt and mt peptide and EHMT2 SET domain in the presence or absence of BIX 01294

CHX treatment also reduced NSD3 protein levels in 30 min indicating short half-life of NSD3 protein in 293s (Fig S8F). To determine whether proteasome activity influences the NSD3 protein's lifespan, we investigated the effect of MG132 on the NSD3 half-life. Contrary to CHX treatment, we found that MG132 increased NSD3 and p21 levels in a time-dependent manner. This indicated that inhibiting the ubiquitin–proteasome system by MG132 treatment stabilizes NSD3 levels by extending the protein's half-life (Fig S8G). Taken together, our results indicated NSD3 protein has short lifespan, and its levels can be stabilized and enhanced by antagonizing ubiquitin–proteasome system or by EHMT2-mediated methylation. Based on the above data, we next hypothesized that NSD3 protein levels are kept at low levels in normal H9-hESCs via the ubiquitin–proteasome system. To address this, H9-hESCs were incubated with or without MG132 (10 $\mu m$) for indicated time points (Fig 8D). Enhanced levels of NSD3 isoforms upon MG132 treatment confirmed our hypothesis (Fig 8D). The enhanced levels of NSD3 are known to stabilize c-Myc protein via protein–protein interaction in cancer cells (65). An increase in c-Myc levels upon MG132 treatment in normal hESCs correlated with stabilization of NSD3 and corroborated with increased c-Myc in V-H9-hESCs (Fig 8D).

Next, we investigated that K477 moiety that gets methylated indeed promotes stabilization of NSD3 and antagonizes degradation via the ubiquitin–proteasome system. Towards this, we transfected Flag-tagged NSD3S Wt or K477R mutant IRES GFP plasmids in 293 cells with or without MG132 treatment. Given this is a transient expression system, transfection efficiency control for Wt and mutant was based on internal GFP IRES expression (Fig 8E; left panel). We noticed comparably the higher expression of GFP in the K477R mutant indicating higher transfection efficiency. GAPDH served as a total protein loading control (Fig 8E; left panel). Probing of input lysates for total ubiquitin levels revealed higher levels of ubiquitin signal in MG132 compared with wt- or mutant-expressing constructs alone (Fig 8E; left panel), confirming the ubiquitin–proteasome system–mediated degradation of proteins upon MG132 treatment. Accordingly, we also noticed higher levels of NSD3 (wt or mutant) in MG132-treated cells (Fig 8E; left panel).

Next, 500 $\mu g$ of cell lysate was subjected to immunoprecipitation using anti-Flag antibody and IPed material was probed using anti-Flag and polyubiquitin antibodies. An equal amount of material was IPed in NSD3S Wt-, NSD3S K477R-, and NSD3S K477R + MG132 treated cells (Fig 8E; right panel). We noticed albeit higher levels of IP Flag in NSD3 Wt treated with MG132 lysates (Fig 8E; right panel). Anti-ubiquitin blot demonstrated that the ubiquitin signal was higher in NSD3S mutant lysate compared to NSD3S Wt (Fig 8E; right panel). These data were consistent with the information in Fig 7H wherein we observed that the NSD3S K477R mutant was unstable. Enhanced ubiquitination signal in NSD3S K477R versus NSD3S K477R + MG132

IPed material further attested that NSD3S protein undergoes degradation via ubiquitin–proteasome machinery (Fig 8E; right panel).

We also tested whether EHMT2 indeed methylates NSD3 at K477 residue and prevents it from ubiquitin-mediated degradation. Towards this, we transfected Flag-tagged NSD3S IRES GFP with Flag-EHMT2 or Flag-K477R IRES GFP with EHMT2-GFP constructs. Lysates were probed for transfection efficiency of NSD3S or K477R using anti-GFP antibody and EHMT2 overexpression monitored by anti-Flag/GFP antibody. Although there was higher transfection efficiency in single plasmid (NSD3S Wt or NSD3S Mt.)-transfected cells (Fig 8F; left panel), equal levels of transfection efficiency were achieved in all Flag-EHMT2–co-transfected cells or EHMT2-GFP–transfected cells. A dose-dependent increase in the expression of Flag-EHMT2 (140 KD) or EHMT2-GFP (165 KD) was seen exclusively in co-transfected cells (Fig 8F; left panel and right panel).

The increased expression of EHMT2 increased Flag-NSD3S Wt protein levels (90 KD) (Fig 8F; left panel); however, such an effect was not seen in Flag-NSD3S K477R mutant–transfected cells (Fig 8F; right panel). We also noticed an additional Flag-NSD3S band migrating at higher sizes only in EHMT2-overexpressing cells, for reasons we are not currently clear about (Fig 8F; left panel). These results convincingly demonstrate EHMT2 methylated K477R residue on NSD3 and prevent ubiquitin-mediated degradation.

Methylation of a lysine residue can preclude the addition of an ubiquitin moiety on the same lysine residue and increase protein stability. To address whether K477 is the residue, which is targeted for both methylation and ubiquitination, we performed an experiment, wherein Flag-NSD3S or Flag-NSD3S K477R was overexpressed and supplemented with MG132. An increase in p21 protein upon MG132 addition served as a positive control (Fig S8H). Probing for anti-Flag antibody demonstrated higher levels of NSD3S in Flag-NSD3S or Flag-NSD3S K477R indicative of stabilization of the NSD3 upon inhibition of the ubiquitin–proteasome system (Fig S8H), because we observed recovery of NSD3 mutant protein only upon MG132 treatment indicated that K477 is the site of methylation but not shared for ubiquitination.

Overall, the data from Fig 5 demonstrate the consequence of NSD3 stabilization leading to pluripotent stem cell transformation, and in Fig 8, we provide understanding on mechanisms by which proto-oncogene NSD3 levels are kept in check by the ubiquitin–proteasome system in normal H9-hESCs.

## Discussion

Preventing the commonly arising variant hPSCs from dominating wild-type cultures necessitates a comprehensive understanding of

---

(n = 2). The H3 peptide was used as a positive control (n = 2). Technical replicates were plotted with mean values. There is a significant difference between wt, mt, & BIX, $P < 0.001$ (Dunn's multiple comparison test). The error bar represents ± SD. **(C, D)** 2D natural abundance $^1H$–$^{13}C$ correlation spectra of time-resolved NMR kinetics for methylation activity of EHMT2 SET on NSD3 K477 peptide. EHMT2 SET was independently able to mono/di/tri-methylate NSD3 at K477. **(E)** Immunoprecipitation with anti-mono-methylated lysine-specific antibodies and immunoblotting for NSD3 in 293s. The black dotted line represents input of the different exposure of the same gel. **(F, G)** Immunoprecipitation with NSD3 antibody and immunoblotting for anti-mono- and anti-di-methylated lysine–specific antibodies in 293s. **(H)** 293s were transfected with indicated plasmids, and immunoblotted with anti-Flag antibody and for anti-GFP antibody. **(I)** Quantification of NSD3S levels in wild-type versus mutant NSD3S-transfected cell lysates.

**Figure 8. NSD3 is regulated by protein degradation mechanisms in normal human ES cells.**
**(A)** Western blot analysis of endogenous NSD3L and NSD3S levels in 293s upon the overexpression of Flag-EHMT2. **(B)** Western blot analysis of endogenous NSD3L levels in 293s upon knockdown of EHMT1 and EHMT2. **(C)** Western blot analysis of endogenous NSD3L levels in 293s upon inhibition of EHMT1/2 with BIX and UNC. **(D)** MG132-

the factors that enable variant cells to achieve clonal dominance. Previous studies have consistently highlighted recurrently acquired aneuploidies and genetic alterations, such as point mutations, contributing to this phenomenon (71, 72). In our study, we demonstrated the dominance of specific variants in hPSC cultures and characterized the dysregulated epigenetic machinery such as EHMT1, EHMT2, and NSDs (NSD1, NSD2, and NSD3). The dysregulation of epigenetic factors (EHMTs and NSDs), combined with EMT, increased cell proliferation and migration, which resembles a previously described progression towards an oncogenic pathway. Among these changes, the up-regulation of NSD3 emerges as a key factor for acquiring the variant phenotype. Initially, we identified differences in the NSD family (NSD1, NSD2, and NSD3) of proteins between wild-type and rapidly growing variant counterparts. Interestingly, NSD2 was not detectable in H9-hESCs and was induced in V-H9-hESCs. NSD2 and NSD3 are both implicated in cancers (52). Among them, proto-oncogene NSD3 is known to transcriptionally regulate cell proliferation genes and stabilize c-Myc via protein–protein interaction (52, 66). Because variants demonstrated enhanced c-Myc and accelerated cell proliferation, we decided to look at role of NSD3 in V-H9-hESCs. Subsequent NSD3 knockdown in variant cells reversed the morphological EMT phenotype completely, rescued cellular proliferation, migration, and differentiation defects largely, and reduced c-Myc levels. Contrary, the overexpression of NSD3 in wild-type cells led to the up-regulation of mesenchymal markers and enhanced proliferation. Partial rescue in growth rates, migration, and differentiation defects in vhESC$^{NSD3-KD}$ despite reduction of NSD3 and c-Myc indicates that this can be attributed to several factors including elevated NSD2 and the overexpression of BCL2L2 and Bcl-XL (10) supplemented with the genetic anomalies in V-H9-hESCs. Nonetheless, our results indicate that NSD3 overexpression in V-H9-hESCs is largely responsible for above-mentioned defects. In addition, our findings also suggest that NSD3 does not act as a stemness factor; rather, maintaining lower levels of NSD3 allows wild-type cells to regulate proliferation and maintain an epithelial state.

Lysine methyltransferases are canonically histone methylators, many of them have overlapping function as non-histone methylators (73). The addition of methyl groups to lysine residues has been shown to impact protein–protein interactions, protein–DNA interactions, protein localization, and particularly protein stability (73). Several non-histone proteins implicated in human cancers have been discovered to be methylated at lysine residues (74, 75). Specifically, lysine methylation on p53, ERa, and CFTR is prevented from proteasomal degradation and is stabilized by methyl modifications on these proteins (76, 77). Using cycloheximide and MG132 treatment experiments, we discovered NSD3 is short-lived protein and subjected to degradation via the ubiquitin–proteasome system. Using the time-resolved NMR, in vitro methyltransferase assay, and domain deletion experiments, we identify NSD3 is subjected to methylation at lysine 477 (K477). Interestingly, lysine 477 is a part of a histone-like ARKS motif, which when mutated leads to degradation of both NSD3S and NSD3L. Our study highlights the importance of EHMT2 and NSD3 interaction, resultant methylation of NSD3, and its stability, which promotes processes such as EMT, survival, and differentiation defects in human pluripotent stem cells. We demonstrate that both EHMT1 and EHMT2 interacted with NSD3 (long and short) and methylated it in in vitro assays (Figs 4F and 7A–D and S7C–H); however, EHMT2 was the only methyltransferase that methylated NSD3 in vivo (Fig 4C–E). Our data strongly support the central role of NSD3 isoforms in altering the fate of pluripotent cells towards a variant state, and methylation on NSD3 as an oncogenic driver.

Mutations, genetic translocations, and altered gene expression involving lysine methyltransferases are frequently observed in cancers and other pathologies (78). Both the isoforms of NSD3 (NSD3L and NSD3S) are linked to cancer. NSD3L has catalytic activity that mediates H3K36 methylation, activating gene transcription (79). For example, NSD3L is implicated in the activation of notch receptors, notch ligands, ADAMs, and reduction in E-cadherin in breast cancers (28). Consistent with the previous studies in breast cancer, our transcriptome analysis revealed up-regulation of notch receptors and ADAMs in variant hES cells. Furthermore, NSD3 overexpression induced EMT mimicking morphological changes, accompanied by loss of epithelial marker E-cadherin and gain of mesenchymal markers, vimentin and N-cadherin in V-H9-hESCs. Contrary to NSD3L, NSD3S contains first PWWP domain, which has the function of binding to H3K36 marks on the chromatin (79). Over the past couple of years, NSD3S has been established as an adaptor protein of important drivers of cancer, such as *MYC*, *BRD4*, and *CHD8* (55, 79). NSD3S has been shown to associate and stabilize MYC protein, thereby increasing its transcriptional activity. Although human ES cells express high levels of c-Myc, variant hESCs demonstrated a further increase in c-Myc levels, possibly inducing an oncogenic phenotype (80).

In this study, we also identified EHMT2 as an upstream regulator for NSD3. Accordingly, the overexpression EHMT2 increases NSD3 levels and depletion of EHMT2 reduces NSD3 levels. Nevertheless, the specific signals upstream of EHMT2 that convey a fitness advantage to variants over wild-type cells during fate transition remain unknown. Our study suggests that differences in epigenetic properties may serve as potential indicators of hPSC cellular fitness. In conclusion, our work reveals the significance of proteostasis of lysine methyltransferases, specifically NSD3 in wild-type H9-hESCs, and that its dysregulation leads to the emergence of a variant state. Further in-depth investigations of the numerous variant lines generated worldwide are warranted to gain a comprehensive understanding of molecular mechanisms (e.g., NSD3 stabilization) impacting hESC transformation. Such studies will be eventually useful for quality control of pluripotent stem cells and possibly designing strategies to prevent hESC transformation.

---

treated H9-ESCs immunoblotted for NSD3L and NSD3S and c-Myc. **(E)** Western blot for immunoprecipitation of 293s transfected with NSD3S Flag and NSD3S K477R treated with and without MG132. The lysates were immunoprecipitated with anti-Flag antibody and immunoblotted for Flag and ubiquitin antibodies. GFP was used as a transfection control. GAPDH was used as a protein control. **(F)** Western blot for NSD3S Flag and NSD3 K477R transfected with EHMT2-GFP in a dose-dependent manner in 293s. Lysates were immunoblotted for Flag and GFP (EHMT) antibodies. GAPDH was used as a protein control. GFP (IRES) was used as a transfection control.

# Materials and Methods

## Cell culture, treatment, and transfections

HEK293 were cultured in DMEM (10566-016, GlutaMAX [Thermo Fisher Scientific]) along with 10% heat-inactivated FBS (10082147; Thermo Fisher Scientific), 2 mM L-glutamine (25030081; Thermo Fisher Scientific), and non-essential amino acids (MEM-NEAA; Thermo Fisher Scientific). The cells were incubated in a 5% $CO_2$ incubator at 37°C. Plasmids were transfected using Xfect reagent (Cat. No. 631318; Takara) according to the manufacturer's instructions. 10 $\mu$m of MG132 was used for treatment on the second day post-transfection. 20 $\mu$g/ml of CHX was used for the treatment. UNC and BIX were used up to 5 $\mu$m for the treatment on 293 cell lines at 70% confluency; 24 h post-treatment, the cells were harvested for the Western blotting and RNA isolation. DMSO was used as a vehicle control for the treatments.

## Human ES cultures and differentiation

H9-hESC, V-H9-hESCs, and V-H9-hESCs[NSD3-KD] were culture in mTeSR media (StemCell Technologies) on hESC grade Matrigel (Corning Matrigel hESC-Qualified Matrix; 354277) and tested by array comparative genomic hybridization (Genotypic Ltd). After 3 d (V-H9-hESCs) or 5 d (hESCs), cultures were dissociated for 1–2 min in ReLeSR (StemCell Technologies) and passed 1:6 (V-H9-hESCs) or 1:3 (hESCs).

For neuronal differentiation, hESCs were dissociated for 5 min in ReLeSR (StemCell Technologies) and replated as aggregates onto poly-L-lysine/laminin-coated plates (BD Biosciences) in neural proliferation media (DMEM/F12 supplemented with 1% N2 [Gibco], 1% B27 [Gibco], 20 ng/ml EGF [R&D Systems], and 20 ng/ml bFGF). After 5–7 d, cultures were dissociated into single cells with Accutase (Sigma-Aldrich). Dissociated single cells were plated back onto laminin-coated plates in neural proliferation media. Media were changed every 3 d. When the cells reached confluence, neural precursors were analysed for the NSC marker nestin.

For astroglial differentiation, cells were first differentiated into neural precursor cells (NSCs). H9-hESCs, V-H9-hESCs, and V-H9-hESCs[NSD3-KD] were dissociated in ReLeSR (StemCell Technologies) for 1–2 min and then replated onto poly-L-ornithine/laminin plates (354659; Corning) in neural proliferation media (DMEM/F12 supplemented with 1% N2 [Gibco], 1% B27 [Gibco], 20 ng/ml EGF [R&D Systems], and 20 ng/ml bFGF). Media were changed at alternate days. After 7 d, NPCs were dissociated into single cells with Accutase (Sigma-Aldrich). Dissociated single cells were then plated back onto laminin-coated plates for glial differentiation in glial media (DMEM/F12 supplemented with 1% N2 [Gibco], 1% B27 [Gibco], 10 ng/ml BMP4 [R&D Biosciences]). Cells were allowed to differentiate for 8 d.

For cardiac differentiation, H9-hESCs or V-H9-hESCs (at 70% confluence) were cultured for in cardiac differentiation medium, IMDM (12-440-053; Gibco), GlutaMax (25030-081; Gibco), L-carnitine (A17618.09; Gibco), non-essential amino acids (11140050; Gibco), b-mercaptoethanol (21985-023; Sigma-Aldrich), and human serum albumin (A9511; Sigma-Aldrich) for 24 h. Then, H9-hESCs or V-H9-hESCs were enzymatically detached and transferred into 0.1%

gelatine-coated petri dishes. The cells were cultured in cardiac differentiation medium with 4 $\mu$m CHIR (Axon) and 2 $\mu$m BIO (Calbiochem) for 24 h to form aggregates. On days 3–9, 10 $\mu$m KY02111 (Tocris Bioscience) was added to cell cultures, and the medium was changed every 48 h. On day 15, the cardiac colonies were detached using protease solution without collagenase, and were transferred to ultra-low-attachment plates (six-well plate, 07-200-601; Corning) with cardiac differentiation medium. Beating colonies were observed after 23–25 d.

## Cell proliferation assay

An equal number of H9-hESCs, V-H9-hESCs, and V-H9-hESCs-NSD3 KD cells were seeded on Matrigel-coated 12-well plates independently in triplicates for each time point. Cell counts were determined for each time point by harvesting the cells using 0.5 mM EDTA. Cell count was later plotted against the specific time points to determine the proliferation rate.

## Scratch assay

H9-hESCs, V-H9-hESCs, and V-H9-hESCs[NSD3-KD] were seeded on Matrigel-coated 24-well plates independently in triplicates. Once the cells reached 70–80% confluence, three scratches were made with a 200-$\mu$l tip vertically in each well carefully under the microscope. The scratches were observed at 0, 16, 24, 36, 48, and 72 h depending on the time the scratch healed completely. Images were taken at each time point. The area of the scratch was calculated using MRI Wound Healing tool Fiji ImageJ (https://github.com/MontpellierRessourcesImagerie/imagej_macros_and_scripts/wiki/WoundHealing-Tool). The scratch area percentage was later plotted against the specific time points to determine the closure rate.

## Generation of NSD3 gene knockdown using CRISPR-Cas9

### CRISPR guide RNA design
The CRISPR guide RNAs for *NSD3* targets were designed using the CRISPOR web tool (http://crispor.gi.ucsc.edu/). The guides were selected based on the high on-target efficiency with minimal off-target sites, especially at the coding regions. Dual guides were designed targeting the intron 1 (5'-CGGGCCTAAAATTATAAAAG-3') and intron 2 (5'-GCCTGAAAGTCGAGTCTTGG-3') to delete the exons of *NSD3* containing the start codon, resulting in 2,880-bp genomic deletion.

## Plasmids and cloning

The sgRNAs were cloned in the PX458-SpCas9-T2A-EGFP vector containing BbsI and BsaI sites by following the reported cloning method with some modifications (81). The guide RNA1 and guide RNA2 were sequentially cloned at the BbsI and Bsa1 sites in the vector, respectively. The cloning was confirmed with Sanger sequencing.

## NSD3 knockdown generation and genotyping

The PX458-SpCas9-T2A-EGFP vector containing the dual-guide RNAs was transfected in 293 cells or V-H9-hESCs with Xfect transfection

reagent (Cat. No. 631317; Takara) or using the neon transfection system as per the manufacturer's instructions. 48 h post-transfection (293) or 4 d post-transfection (V-H9-hESCs), cells were harvested, and GFP-positive cells were bulk-sorted using BD FACSMelody Cell Sorter and replated for further amplification/colony picking for PCR screening. For validating the genomic deletion, gDNA was isolated from bulk-sorted cells by Wizard Genomic DNA Purification Kit (Cat. No. A1120; Promega Corporation) followed by PCR genotyping using primers flanking the deletion site (FP: 5′-TTGATTAGCCTCTAGCCTATGAG-3′; RP: 5′-CAAAATTCAATTAACAAGAACTTACCTC-3′). The 328-bp PCR product from the gDNA of transfected cells validated the deletion of *NSD3* exons in contrast to the 3,134-bp PCR product from that of control cells. A V-H9-hESC$^{NSD3-KD}$ cell line that was generated from bulk-sorted cells was cultured and maintained in mTeSR medium.

### Antibodies, inhibitors, and reagents

The following antibodies were used in the current study: NSD3 (180500, rabbit monoclonal; Abcam), NSD3 (11345-1-AP, rabbit polyclonal; Protein Tech), EHMT1(A301-642A, rabbit polyclonal; Bethyl Laboratories), EHMT2(NBP2-13948, rabbit polyclonal; Novus Biologicals), EZH2 (07689, rabbit polyclonal; MERCK Millipore), E-cadherin (ab231303, mouse monoclonal; Abcam), E-cadherin (ab15148, rabbit polyclonal; Abcam), vimentin (ab92547, rabbit monoclonal; Abcam), snail (ab53519; Abcam), beta-catenin (ab16051, rabbit polyclonal; Abcam), nestin (ab22035, mouse monoclonal; Abcam), H3K27ac (ab4729, rabbit polyclonal; Abcam), H3K9me2 (ab1220, mouse monoclonal; Abcam), H3K36me2 (ab9049, rabbit polyclonal; Abcam), H3K27me3 (07-449, rabbit polyclonal; Millipore), H3 (ab1791, rabbit polyclonal; Abcam), Flag (ITM3001, mouse monoclonal; ImmunoTag), GAPDH (G9545, rabbit polyclonal; Sigma-Aldrich), ubiquitin (BML-Pw0930-0100, mouse; Enzo Life Sciences), anti-methyl-lysine antibody (ICP0501, rabbit polyclonal; ImmuneChem), p21 (sc6246, mouse monoclonal; Santa Cruz), anti-6X His-tag antibody (ab9108, rabbit polyclonal; Abcam), mono-methylated lysine (14679S, rabbit monoclonal; Cell Signalling Technologies), anti-GFP (ab290, rabbit polyclonal; Abcam), normal rabbit IgG (12-370; Millipore), and normal mouse IgG (12-371; Millipore). Secondary antibodies goat anti-mouse HRP (172-1011) and goat anti-rabbit HRP (170-8241) were purchased from Bio-Rad. MG132 (C2211), CHX (C4819), BIX01294 (B9311), and UNC0642 (SML1037) inhibitors were purchased from Sigma-Aldrich.

### RNA extraction

Cells were harvested, washed in 1X PBS, and then resuspended in TRIzol for total RNA extraction. The conventional method of chloroform for phase separation and isopropanol for precipitation of RNA was used. cDNA was obtained from RNA by performing reverse transcription using PrimeScript RT Reagent Kit (RR037A; Takara). Quantitative RT–PCR was then performed on the cDNA using TB Green Premix Ex Taq II (RR820A; Takara).

### Quantitative PCR

Total RNA was isolated from the cells using TRIzol reagent (Cat. No. 15596026; Invitrogen) as per the manufacturer's instructions. cDNA was synthesized from RNA using PrimeScript RT Reagent Kit (RR037A; Takara).

RT–PCR was performed from cDNA using TB Green Premix Ex Taq II (RR820A; Takara) as per the manufacturer's instructions. The following are the primers used for qRT-PCR.

**List of Primers used for qRT-PCR.**

| NSD3 Short qRT-PCR FP | 5′-CGCACAAATTCCACATAGAG-3′ |
|---|---|
| NSD3 Short qRT-PCR RP | 5′-GGTTGCAGAGACATCTCCATC-3′ |
| NSD3 Long qRT-PCR FP | 5′-CAGGCTACAGTGAAGACTGG-3′ |
| NSD3 Long qRT-PCR RP | 5′-CAGGAAGTGATGCCAATCC-3′ |
| beta actin F | 5′-GCGTACAGGGATAGCACAGC-3′ |
| beta actin R | 5′-GGATTCCTATGTGGGCGACGA-3′ |
| EHMT1FP | 5′-GACATCAACATCCGAGACAA-3′ |
| EHMT1RP | 5′-GAAAGAAAGAGGACGACACAG-3′ |
| EHMT2FP | 5′-GCCATGCCACAAAGTCATTC-3′ |
| EHMT2RP | 5′-CTCAGTAGCCTCATAGCCAAAC-3′ |
| GAPDHFP | 5′-GAAATCCCATCACCAATCTTCCAGG-3′ |
| GAPDHRP | 5′-GCAATTGAGCCCCAGCCTTCTC-3′ |
| ACTN2 F | 5′-GGCGTGCAGTACAACTACGTG-3′ |
| ACTN2 R | 5′-AGTCAATGAGGTCAGGCCGGT-3′ |
| Myh62 F | 5′-GGGGACAGTGGTAAAAGCAA-3′ |
| Myh62 R | 5′-TCCCTGCGGGCCACTATCTT-3′ |
| TNNT29 F | 5′-TTCACCAAAGATCTGCTCCTCGCT-3′ |
| TNNT29 R | 5′-TTATTACTGGTGTGGAGTGGGTGTGG-3′ |
| BRACHYURY F | 5′-CGGAACAATTCTCCAACCTATT-3′ |
| BRACHYURY R | 5′-GTACTGGCTGTCCACGATGTCT-3′ |
| MESP 1 F | 5′-CTCGTCTCGTCCCCAGACT-3′ |
| MESP 1 R | 5′-AGCGTGCGCATGCGCAGTT-3′ |
| TBX6 F | 5′-AGGCCCGCTACTTGTTTCTTCTGG-3′ |
| TBX6 R | 5′-TGGCTGCATAGTTGGGTGGCTCTC-3′ |
| ISL1 F | 5′-CATCGAGTGTTTCCGCTGTGTAG-3′ |
| ISL1 R | 5′-GTGGTCTTCTCCGGCTGCTTGTGG-3′ |
| SLITRK1 F | 5′-TGCTCCAAGACCTCCTCATA-3′ |
| SLITRK1 R | 5′-GCCCAGATCCAACAGAATGA-3′ |
| SLITRK5 F | 5′-CAAACCTCCAGCTGCTATTCT-3′ |
| SLITRK5 R | 5′-CAAGGAGGTGAAGTGGTTACTC-3′ |
| ROBO2 F | 5′-GGATGAGTTGACAAGAGCCTATC-3′ |
| ROBO2 R | 5′-GGCTCCAGACACATAACCTAAC-3′ |
| E CADHERIN F | 5′-ATTCTGATTCTGCTGCTCTT-3′ |
| E CADHERIN R | 5′-AGTAGTCATAGTCCTGGTCTT-3′ |
| OCT4 F | 5′-TTAGAAGGCAGATAGAGCCACTGACC-3′ |
| OCT4 R | 5′-TGCCTGTCTGTGAGGGATGATGTT-3′ |
| Human PAX6 FP | 5′-CAGAGAAGACACAGGCCAGCAA-3′ |
| Human PAX6 RP | 5′-AGCGCTGTAGGTGTTTGTGA-3′ |
| GFAP FP | 5′-AGGATGGAGAGGAGACGGCAT-3′ |
| GFAP RP | 5′-GGATGGAGAGGAGACGCATCA-3′ |
| NF1A_pp1_FP | 5′-ACCGATTCACCATGGAGGGC-3′ |
| NF1A_pp1_RP | 5′-AGTGGCAACTGATGAGCAAGC-3′ |

### Western blotting

The whole-cell lysate was prepared by suspending the cell pellet in RIPA buffer supplemented with a 1X concentration of protease inhibitor cocktail, followed by incubation on ice for 1 h with intermittent vortexing. Protein concentrations of the lysates were determined using the Bradford assay. Subsequently, specific amounts of protein were treated for reduction and denaturation by combining with 10X NuPAGE Sample Reducing Agent and 4X NuPAGE LDS sample loading buffer, followed by heating at 70°C for 10 min. These treated samples were then separated on a 10% SDS–PAGE gel and subjected to immunoblotting using appropriate antibodies. The resulting bands on the Western blot were analysed and quantified using ImageJ software.

### Immunoprecipitation

Protein A or G Dynabeads underwent two washes with 0.1% BSA in 1 M potassium phosphate buffer. After this, 2 µg of the respective antibody, diluted in 0.5 M potassium phosphate buffer, was introduced and allowed to incubate on a rotor for 2 h at room temperature. The excess, unbound antibody was subsequently removed through washing with potassium phosphate buffer, after which the protein lysate was added and left to incubate overnight at 4°C. The next day, any unbound fraction was eliminated by washing with IP-100 and 1X Flag buffer. The immunoprecipitated protein was then eluted by subjecting it to an incubation with 2X reducing agent and protein loading dye at 70°C for 10 min.

### Cloning and site-directed mutagenesis

The NSD3-long isoform that was amplified from human NSD3-long isoform in pcDNA3.1 plasmid GenScript clone (Clone ID: G32253) was a gift from Dr. ChandraPrakash Chaturvedi. The amplified product was cloned in EGFPC1 (6084-1; Clontech) between BamH1 and Sal1 restriction sites. The EHMT2 full-length gene was amplified from Flag-msEHMT2-pcDNA5 (gift from Dr. Marjorie brand). The amplified msEHMT2 full length was cloned in EGFPC1 between Kpn1 and BamH1 restriction sites.

Primers used for cloning NSD3L in EGFPC1 were as follows: NSD3FL_F-5′-ACGCGTCGACAAATGGATTTCTCTTTCTCTTTCATG-3′; NSD3FL_R-5′-CGCGGATCCTTATTCTTTTACTTCTTCTCCATGATC-3′.

Primers used for cloning EHMT2 in EGFPC1 were as follows: EGFPC1_EHMT2_KPN1_F-5′-AAAAGGTACCATGCGGGGTCTGCCGAG-3′; EGFPC1_EHMT2_BAMH1_R-5′-TTTTGGATCCTTGGTGTTGATGGGGGGC-3′.

To clone NSD3S in an 8xHis-tag vector, NSD3-short isoform was digested from NSD3-short MSCV (#72552; Addgene) using Nco1 and EcoR5 enzymes. The gel-purified fragment was cloned in pTriEx-4 Neo (Cat. No. 70933-3; Sigma-Aldrich) between Nco1 and Pvu2 restriction sites.

To clone NSD3 1-384aa, NSD3-short MSCV (#72552; Addgene) was digested with Nco1 and Xho1. The fragments were resolved in agarose gel, and 1146 fragment containing NSD3 1–384 aa was cloned into Nco1- and Xho1-digested pTriEx-4 Neo.

To clone NSD3 1–173 aa, NSD3-short isoform was digested from NSD3-short MSCV (#72552; Addgene) using the BSA1 enzyme, 2,352 fragments were gel-purified, and the ends were blunted using

Phusion polymerase and later digested with Nco1. The fragment containing NSD3 1–173 aa was cloned into Nco1 and Pvu2 sites of pTriEx-4 Neo. All the clones were sequenced to confirm the identity.

### Site-directed mutagenesis

Site-directed mutagenesis was carried out using Phusion polymerase with the following program: initial denaturation at 98°C for 1 min, 18 cycles (98°C for 5 s, 72°C for 5 s, 72°C for 2 min), and final extension at 72°C for 5 min. The reaction was later digested using 10 U of Dpn1 (R0176; New England Biolabs) enzyme at 37°C for 1 h. The final product was transformed into E.coli DH5α competent cells. The plasmid was isolated using Promega (Wizard Plus SV Minipreps DNA Purification System). The following primers were used for the site-directed mutagenesis.

**Primers used for generation of NSD3 mutant constructs.**

| NSD3 K477A F | 5′-GAAGCTGGTAAGGATGCCCTTGCTGCCGCAG-3′ |
| --- | --- |
| NSD3 K477A R | 5′-CTGCGGCAGCAAGGGCATCCTTACCAGCTTC-3′ |
| NSD3 K477R F | 5′-GAAGCTGGTAAGGATCTCCTTGCTGCCGCAG-3′ |
| NSD3 K477R R | 5′-CTGCGGCAGCAAGGAGATCCTTACCAGCTTC-3′ |
| NSD3ΔPWWP-F | 5′-GAAGTACAGGCAAGTGAGGCTTTATCCCAAGCAAAAAAG-3′ |
| NSD3ΔPWWP-R | 5′-CTCTTTTTTGCTTGGGATAAAGCCTCACTTGCCTGTAC-3′ |

### Protein purification

The human 6xHis-EHMT1 SET and 6xHis-EHMT2 SET domain–containing plasmids (#25504 and #25503; Addgene, respectively) were a gift from Cheryl Arrowsmith. EHMT1 and EHMT2 SET domain–containing plasmids were expressed in E.coli C43 (DE3), whereas 8xHis-NSD3-short plasmid was expressed in E.coli BL21(DE3). E.coli containing EHMT1 and EHMT2 SET domain–containing plasmids were grown in LB medium with 50 µg/ml of kanamycin, whereas E.coli BL21(DE3) with 8xHis-NSD3-short was grown in LB medium with 100 µg/ml of ampicillin. Bacteria were induced with 250 µm isopropylthiogalactoside when OD600 was 0.6–0.8. After induction, the cell pellet was resuspended in lysis buffer containing 25 mM Tris (pH of 8.0), 500 mM NaCl, 5% glycerol, 0.05% Nonidet P-40 substitute (M158; Amresco), 0.2 mM PMSF (Cat. No. 93482; Sigma-Aldrich), and 30 mM imidazole. Cells were lysed by sonication (Sonics & Materials Inc.). The lysate was centrifuged, and the supernatant was incubated o/n with 2 ml of Ni-NTA agarose resin equilibrated with lysis buffer described above. The resin was then packed into Econo-Column (738-0014; Bio-Rad). The resin column was washed, and His-tagged proteins were eluted from the resin using 100 and 250 mM imidazole. The protein was then dialysed using snakeskin dialysis membrane and concentrated using Amicon Ultra-4 centrifugal concentrators (Millipore) with the cut-off of 3 KD. The purity was finally assessed using SDS–PAGE.

### In vitro methyltransferase assay

#### Detection by fluorescence

The methyltransferase assay kit (ADI-907-032; Enzo Life Sciences) was used to perform the in vitro methyltransferase assay. The assay was performed as per the manufacturer's instructions using the His-tag purified EHMT2 SET domain. The NSD3 21 aa peptide was synthesized from Pepmic Co., Ltd. The NSD3 wt peptide sequence is VKIAWKTAAARKSLPASITM, and the NSD3 K477A mt peptide sequence is VKIAWKTAAARASLPASITM.

#### Detection by Western blotting

8xHis-NSD3-short was incubated with different concentrations of EHMT2 SET along with 40 $\mu$m SAM in a 30 $\mu$l reaction. The reaction was incubated overnight at 37°C, and the reaction was stopped by adding 4x SDS loading dye along with 10x reducing agent. Samples were heated at 95°C for 5 min, resolved in 8% SDS–PAGE, and proceeded for Western blotting with anti-methylated lysine antibody.

#### Immunocytochemistry

Human ES cells were seeded on glass coverslips. 24 h post-seeding, cells were fixed with 4% paraformaldehyde (wt/vol) for 10 min at RT followed by 3 1XPBS washes and permeabilization with 0.5% Triton X-100 (vol/vol) in 1% BSA for 10 min at RT. Primary antibodies diluted in 5% BSA at concentrations indicated below were incubated overnight at 4°C. For immunolabelling, E-cadherin (1:100), vimentin (1:500), $\beta$-catenin (1:200), and snail (1:200) antibody dilutions were used. The samples were incubated overnight at 4°C, followed by PBS washes. The Alexa secondary fluorophore-conjugated antibodies were mixed at a dilution of 1:200 and were incubated for 40 min at RT. DAPI was added for 5 min before mounting. Air-dried coverslips were mounted on glass slides with Vectashield mounting media. Confocal imaging was performed on the Leica SP8 confocal system. Image analysis and extraction of raw files were done with ImageJ.

#### NMR spectroscopy

All NMR spectra were recorded at 303 K on an 800-MHz Bruker Avance III spectrometer. 2D natural abundance $^1$H–$^{13}$C correlation NMR spectra were acquired using standard heteronuclear single quantum coherence NMR experiments. NSD3 21–amino acid peptide containing histone-like ARKS motif was commercially synthesized along with the lysine-to-alanine mutant from Pepmic Co., Ltd. Methyltransferase reaction was carried out in deuterated 25 mM Tris/40 mM NaCl buffer of pH 7.5 along with 2 mM SAM (purchased from Sigma-Aldrich) along with 1 mM peptide and 2 $\mu$m of purified EHMT SET in a NMR tube of 3 mM diameter (1). H/$^{13}$C heteronuclear single quantum coherence spectra were acquired at various intervals from 0, 4, 8, 12, and 16 h. 6x His-tagged EHMT1 and EHMT2 SET domains used in the methyltransferase reaction were purified from E.coli CD43DE3 using Ni-NTA agarose beads followed by size-exclusion chromatography.

### Bulk RNA-sequencing analysis

In this study, a total of six hESC samples were analysed, including three control samples (H9-hESC) and three variant samples (V-H9-hESC).

#### Quality control and preprocessing of raw reads

Initial raw sequencing data are subjected to quality control (QC) using FastQC (version 0.11.9). This analysis assessed base quality score distributions, sequence quality scores, and GC content distribution. Subsequently, adaptors and low-quality bases are processed using Trim Galore (version 0.6.6), which incorporates Cutadapt (version 3.4) and FastQC, thus optimizing the trimming process with its default cut-off parameters.

#### Alignment and differential gene expression analysis

Furthermore, the processed reads are aligned to the human reference genome (GRCh38) using the HISAT2 (v2.2.1) alignment tool with its default settings. After alignment, the generated SAM files are converted into BAM files using SAM tools (v1.10) followed by sorting and indexing. Finally, htseq-count is used to produce the raw count matrices using BAM files.

Differential gene expression analysis between the control (H9-hESC) and variant samples (V-H9-hESC) is performed using the DESeq2 (v1.30.1) package. The results are adjusted for multiple testing with the Benjamini–Hochberg method to control the false discovery rate (FDR). We identified differentially expressed genes using an adjusted $P$-value threshold of less than 0.05 and a log$_2$ fold change of at least 0.75.

### Ubiquitin assay

HEK293 were cultured in DMEM (10566-016, GlutaMAX [Thermo Fisher Scientific]) along with 10% heat-inactivated FBS (10082147; Thermo Fisher Scientific), 2 mM L-glutamine (25030081; Thermo Fisher Scientific), and non-essential amino acids (MEM-NEAA; Thermo Fisher Scientific). The cells were incubated in a 5% $CO_2$ incubator at 37°C. NSD3 WT and NSD3 K477R Mt plasmids were transfected using Xfect reagent (Cat. No. 631318; Takara) according to the manufacturer's instructions. 10 $\mu$m of MG132 was used for treatment on the second day post-transfection. The whole-cell lysate was collected 48 h post-transfection as mentioned in the section for Western blot, and immunoprecipitation was done for NSD3 Wt, NSD3 Wt +MG132, NSD3 Mt, and NSD3 Mt+MG132 samples as mentioned in the Methods section for immunoprecipitation. Blots were probed for ubiquitin antibody by Western blotting.

### NSD3 K477R MT and EHMT2 overexpression

HEK293 were transfected with Flag-NSD3 wild-type or Flag-NSD3 K477R mutant plasmid along with EHMT2 (Flag or GFP tagged) at the indicated concentration using Xfect reagent. The whole-cell lysate was collected after 48 h, and Western blot was done as mentioned in the Methods section for Western blot. The blot was probed for Flag, GAPDH, and GFP antibodies.

### Statistical analysis

The data are represented as the mean ± SD. The graphs were plotted with GraphPad Prism 8; t test, one-sample t test, and one-way ANOVA were used for statistical analysis. We have performed the statistical analysis for H9 and V-H9 for all three biological

replicates (n = 3) in each of the conditions mentioned in the graph of Fig 2D–F. For comparison between H9- versus V-H9-hESCs in each separate condition (ESCs, NSCs, and glial cells), we performed an unpaired $t$ test to test the significance of each of the genes tested: for GFAP, unpaired $t$ test, $*P$ = 0.0165 for H9- and V-H9-ESCs, $**P$ = 0.0035 for H9- versus V-H9-NSCs, and $****P$ < 0.0001 for H9 and V-H9 astroglial cells (error bars: SEM); for NF1A, unpaired $t$ test, ns $P$ = 0.0592 for H9- and V-H9-hESCs, $**P$ = 0.0067 for H9- versus V-H9-NSCs, and $****P$ < 0.0001 for H9 and V-H9 astroglial cells (error bars: SEM); for PAX6, unpaired $t$ test, ns $P$ = 0.0938 for H9- and V-H9-ESCs, ns $P$ = 0.3761 for H9- versus V-H9-NSCs, and $****P$ < 0.0001 for H9 and V-H9 astroglial cells (error bars: SEM). Statistical analysis was performed for H9- and V-h9-hESCs separately and for H9- and V-H9-hESC$^{NSD3-KD}$. This, however, is not represented in the figures included in the article (n = 3); for GFAP, unpaired $t$ test, $****$ $P$ ≤ 0.0001 for H9- and V-H9-ESCs, and $**P$ = 0.0023 for H9- and V-H9-hESC$^{NSD3-KD}$ (error bars: SEM); for NF1A, unpaired $t$ test, $****$ $P$ ≤ 0.0001 for H9- and V-H9-hESC, and $**P$ = 0.0085 for h9 versus V-H9-hESC$^{NSD3-KD}$ (error bars: SEM); and for PAX6, unpaired $t$ test, $**P$ = 0.0059 for H9- and V-H9-hESCs, and $**P$ = 0.0035 for H9- and V-H9-hESC$^{NSD3-KD}$ (error bars: SEM).

# Supplementary Information

# Acknowledgements

This project was supported by funds from DST-SERB (EMR/2016/007076) to S Rampalli, funds from CSIR-IGIB (MLP 2008) to S Rampalli, and inStem Core Funds to S Rampalli. VK Krishnamoorthy and KK Biligiri are supported by CSIR-JRF/SRF fellowship. We thank Prof. Colin Jamora and Dr. Sunil Laxman for their critical and constructive feedback on the article. We thank Dr. Debojyoti Chakraborty and his laboratory for human ES work and CRISPR-Cas9-related experiments at CSIR-IGIB. The Central Imaging and Flow Cytometry Facility (CIFF) at InStem and NCBS and Confocal Microscopy Facility at CSIR-IGIB supported the confocal microscopic imaging. The NMR data were acquired at the NCBS-TIFR NMR Facility, supported by the Department of Atomic Energy, Government of India.

## Author Contributions

VK Krishnamoorthy: conceptualization, data curation, formal analysis, validation, and writing—original draft, review, and editing.
F Hamdani: data curation, formal analysis, validation, visualization, project administration, and writing—original draft, review, and editing.
P Shukla: data curation, formal analysis, validation, visualization, and writing—original draft.
RA Rao: data curation and formal analysis.
S Anaitullah: data curation, formal analysis, and validation.
KK Biligiri: data curation, visualization, and writing—review and editing.
RV Kadumuri: formal analysis.
PR Pothula: formal analysis.
S Chavali: formal analysis, supervision, validation, and writing—review and editing.
S Rampalli: conceptualization, resources, data curation, formal analysis, supervision, funding acquisition, validation, visualization, methodology, data analysis, and writing—original draft, review, and editing.

## Conflict of Interest Statement

The authors declare that they have no conflict of interest.

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
