## [Reviewer comments · Life Science Alliance]

Life Science Alliance

NSD3 protein methylation and stabilization transforms human ES cells into variant state

Shravanti Rampalli, 3. Vignesh Krishnamoorthy, Fariha Hamdani, Pooja Shukla, Radhika Rao, Shaikh Anaitullah, Kriti Biligiri, Rajashekar Kadumuri, Purushotham Pothula, and Sreenivas Chavali

DOI: <https://doi.org/10.26508/lsa.202402871>

Corresponding author(s): *Shravanti Rampalli, Institute of Genomics and Integrative Biology*

Review Timeline:	Submission Date:	2024-06-05
	Editorial Decision:	2024-07-16
	Revision Received:	2024-10-29
	Editorial Decision:	2024-11-20
	Revision Received:	2024-11-29
	Accepted:	2024-11-29

Transaction Report:

July 16, 2024

Re: Life Science Alliance manuscript #LSA-2024-02871-T

Dr. Shravanti Rampalli
CSIR-Institute For Genomics and Integrative Biology (CSIR-IGIB)
DTC Bus Depot, South Campus, Mathura Road, near to Sukhdev Vihar, New Delhi
New Delhi, Delhi 110025
India

Dear Dr. Rampalli,

Thank you for submitting your manuscript entitled "NSD3 protein methylation and stabilization transforms human ES cells into variant state". The manuscript has been evaluated by expert reviewers, whose reports are appended below. Unfortunately, after an assessment of the reviewer feedback, our editorial decision is against publication in Life Science Alliance.

Although your manuscript is intriguing, I feel that the points raised by the reviewers are more substantial than can be addressed in a typical revision period. If you wish to expedite publication of the current data, it may be best to pursue publication at another journal.

Given the interest in the topic, I would be open to re-submission to Life Science Alliance of a significantly revised and extended manuscript that fully addresses the reviewers' concerns and is subject to further peer review. If you would like to resubmit this work to Life Science Alliance, you may submit an appeal directly through our manuscript submission system. Please note that priority and novelty would be reassessed at re-submission.

Regardless of how you choose to proceed, we hope that the comments below will prove constructive as your work progresses.

Thank you for thinking of Life Science Alliance as an appropriate place to publish your work.

Sincerely,

Reviewer #1 (Comments to the Authors (Required)):

Advanced summary:

In their manuscript, Krishnamoorthy and colleagues use their previously isolated variant hESC (vhESC), as a model system to understand oncogenic transformation. They first show that vhESC possess features of mesenchymal cells and an impaired differentiation potential. They next identify genetic alterations which might explain the proliferative and differentiation phenotype of their variant. They also observe that vhESC possesses a higher level of methyltransferases (MT) compared to wt hESC and that NSD3 KO restores epithelial characteristics of vhESC and a normal proliferative rate. They therefore decide to focus the rest of their study on the protein-protein interactions and relevance of MT during transformation using a range of biochemistry approaches. The authors conclude that:

1. EHMT1/2 interact with NSD3
2. EHMT1/2 methylate NSD3 on Lysine residue 477 (K477)
3. NSD3 K477 methylation protects NSD3 from proteasome degradation
4. Elevated NSD3 mediates transformation

Overall this study could in principle provide novel useful information that should be of interest to the audience of the journal. However, I have concerns regarding the quality of the data, their interpretation and the general presentation and clarity of the manuscript. I do not recommend publication at this stage and would advise major revisions.

Major comments:

1) Fig1: The authors claim that their results "demonstrate that V-H9-hESCs display aberrant differentiation features, particularly toward the astrocytic lineage" but the data is weak:

Only one image of a GFAP staining is shown where some signal is clearly visible in the variant cell line. The difference in background intensity may indicate different contrast adjustments between the two conditions. To validate this result, the authors could perform qPCR for relevant markers as they did for the cardiac lineage. Furthermore, it would be interesting to determine if the variant cell line has lost competence to differentiate to the astrocytic lineage or if the differentiation is only delayed / less efficient? Note that there is no indication of the number of days at which the analysis was performed. It would be helpful to include a small diagram depicting the timeline of the treatments and analysis in the figure. If this differentiation phenotype is confirmed, how do the authors explain the phenotype? Is this phenotype rescued with V-H9-NSD3 KO cells?

2) Fig 2H could be an interesting source of information to delineate the genetic changes associated with transformation. Only CDH1 and BCL2L2 are annotated and these genes are not the most significantly differentially regulated. Could the authors include the list of genes as a supplementary table and provide at least a gene ontology analysis in their main figure?

3) Figure 3 is particularly important for the main thesis of the paper and is used to demonstrate that EHMT 1 and 2 can methylate NSD3. I am not convinced by the claim that EHMT1 is able to methylate NSD3. Interactions between EHMT1 and NSD3 S and L seem weak (Fig 3B where we see a faint band compared to the very strong signal in the input - Fig3C, the Flag antibody detects multiple bands, how can the authors be sure they detect a specific product - why not use their NSD3 antibody directly)

While I agree that a higher NSD3S methylation is detected when EMHT2 is overexpressed, I cannot see the same effect when EMHT1 is overexpressed. Please revise in the main text

NB: there is no indication of the number of time this experiment was repeated. Given the importance of the data, I would recommend the authors to show at least 2 replicates.

Fig 3E raises concerns. The image intensity is clipped and gives the impression that information is being hidden. We observe signal in the absence of SAM which is odd. We also observe signal in the absence of EHMT2_SET. Is this an indication that NSD3 may methylate itself, or a technical artifact? There also seems to be a plateau effect (no further increase in methylation beyond 0.5x EHMT2). Please comment on all these points.

Fig 3F seems to be mis-annotated. there are 6 lanes in the image but only 5 are annotated. See the negative bands between Input and IP-IgG
Overall this figure does not inspire confidence and I would recommend that the authors provide raw images with their revision.

4) Fig 5A. The authors claim here that they could detect "a strong interaction of EHMT2 with NSD3L isoform, NSD3 Short isoform and N-term NSD3 (1-384). We observed no interaction of EHMT2 with NSD3 N-term (1-173) domain". However, there are multiple concerning issues with this panel:

- There is no condition probing for EHMT2 and NSD3S interaction in this experiment (lane 7 is only showing that the flag antibody can bind to Flag-NSD3S)
- The image shows lanes with different contrast adjustments. This is not acceptable as this does not allow us to compare lanes with one another. Lane 8 for example has been contrast adjusted but the corresponding input was not. Even if a band is detected, we cannot conclude that there is a strong interaction here.
- Even more problematic, the input for NSD3 1-173 was contrast adjusted to show a very faint band indicating that very little protein was produced in this condition. On the other hand, the corresponding flag pull-down was not adjusted, thus "conveniently" showing no band at all. If the authors wish to maintain their claim, they should repeat this experiment and provide original raw images of the blots.

5) Fig 6A and B present an in vitro assay measuring the methylation of a 21 aa peptide by either EHMT1 or EHMT2. I am a little confused by the approach as the authors mentioned earlier that the PWWP domain was essential for NSD3 and EHMT2 interaction, and therefore methylation. The authors should clarify their rationale and incorporate the sequence and position of the 21 aa peptide in the main figure. (The fact that the peptide contains other lysine residues is an important information for interpretation of the data)

6) Fig 7: The authors make the strong claim that NSD3 K477 methylation protects NSD3 from proteasome degradation. However they do not provide evidence for this. Fig 7E only shows that the addition of the proteasome inhibitor MG 132 leads to a higher level of NSD3 wt and K477R. This only shows that NSD3 is susceptible to proteasome degradation independently of K477. We do not observe any further reduction in the level of K477R compared to the wt in the absence of MG132. Please moderate the claim or perhaps try an anti-ubiquitin antibody against both versions of NSD3 after IP.

Other comments:

* Generally, the quality of the immunoblots is questionable. For example, we see at least two main bands in Fig 1D but this is not commented on. Same for Fig3 C with multiple bands. (IP-v5 - IB: Flag); Fig 4D, the lanes are not well aligned with rows including 3 lanes while others include 2 lanes.

* The molecular weight is often missing from the various immunoblots - the ladder is never annotated.

* Most scale bars are difficult to see (Fig 1A, E - absent in I - Fig 4A,E). The scale bar is given in pixel in F-H

* Fig 2H: The legend appears behind the volcano plot and cannot be read

* fig legends are incomplete - for example Fig 1F-J, what is the time of analysis post induction of differentiation? Number of independent experiments. Fig 3E: the acronyms SAM and SET are not defined; Fig 6C and D, what do the pink and blue color represent...

* Fig1E: There is no comment in the main text regarding the Snail staining in Fig 1E. Furthermore both wt and vHESC are positive for Snail, please comment

* P7 - "Distinct chromosomal abnormalities detected in multiple studies including ours suggest that these cannot be used as specific markers of H9-hESCs transformation." I am not sure I understand what the authors mean here - Please clarify

...

Reviewer #2 (Comments to the Authors (Required)):

In this study, Rampalli et al. investigated the difference between the commonly used human embryonic stem cell line, H9-hESC, and the variant H9-hESC (V-H9-hESC). The V-H9-hESC exhibited dysregulated EMT-specific markers and compromised differentiation. They focused on the epigenetic level changes and found that V-H9-hESC has increased expression levels in EHMT1, EHMT2, and NSD3. They also found that the NSD3 knockout can rescue the EMT marker dysregulation and the phenotypes in V-H9-hESC. To investigate the mechanism further, they found that EHMT1 and EHMT2 can interact with the PWWP domain in NSD3. They performed IP and methylation assay and found that the K477 residue in NSD2 can be methylated and stabilized in the protein from proteasome degradation. Overall, this study provides an interesting hESC line as an EMT study tool and molecular mechanism for the NSD3 lysine residue methylation from the EHMTs. However, there are several major concerns about the data quality, and the experimental result didn't justify the conclusions well. Here are the specific comments and suggestions:

Major comment:

1. In Figure 1, the author showed the morphology changes in the variant H9-hESC line and changed expression level of EMT-related factor compared to H9-hESC. It is critical to quantify or measure the EMT phenotype using a wound healing assay in variant H9-hESC. It is also acquired in NSD3 KO V-H9-hESC in Fig 4 to check the rescue phenotypes.
2. H3K36me2 is significantly increased in the V-H9-hESC line compared to the parental line, however, the level of their methyltransferase NSD3L seems mildly increased in the V-H9-hESC. Is there another H3K36 methyltransferase level changes?
3. In Figure 2, Panel AB: The author found that some genes that are important for cell proliferation and differentiation have small amplifications and deletions on the genome of the V-H9-hESC line. How do they preclude the possibility that the issue causes the EMT phenotype in the V-H9-hESC line?
4. In Figure 5, Panel A: The immunoblotting data are spliced, the authors need to provide the original data (also in Figure 3 A). It is confusing why there is a Flag-NSD3S-only group and the pull-down efficiency is low in the NSD 1-384 group. It is worth replacing the Flag-NSD3S-only group with NSD3S (without Flag) and Flag-EHMT2 co-transfection group.
5. The authors investigated the stabilization of NSD3 as being associated with the methylation of K477 residue and proteasome degradation. Although they found the K477 residue didn't share the ubiquitination site, they still suggested that the NSD3 protein methylation prevented the ubiquitination in the text and Figure 8. The data for this part is insufficient.
6. Some sentences didn't align the data or need more explanation. As in P.7, "Distinct chromosomal abnormalities detected in multiple studies including ours suggest that these cannot be used as specific markers of H9-hESCs transformation." There is no evidence to support this phenomenon. Some sentence also seems redundant. The author needs to modify the description to be more precise in this manuscript.

Minor comment:

1. It needs better image quality (some have shallow resolution), and the figure arrangement (is not consistent) in this manuscript.
2. Better characterization of cell differentiation in Fig1. For example, in Fig1 I, J didn't show the differentiation ability of V-H9-hESC to mesoderm precursor cells.
3. Fig 2 G, H figure issue, and the correlation with epigenetic changes (seems not related to other data)
4. NSD3 KO/KD issue: The knockout cell line also expresses NSD3L and NSD3S (methods : bulk or single colony)
5. What is the FT group in Fig 5 B?
6. OE NSD3 didn't affect EHMT2 level, however, KD NSD3 reduced EHMT2 level
7. In Figure 7, Panel F: The protein stability assay didn't reflect the correlation of NSD3 stabilization and the increased cMyc level.

Reviewer #1

Advanced summary: In their manuscript, Krishnamoorthy and colleagues use their previously isolated variant hESC (vhESC), as a model system to understand oncogenic transformation. They first show that vhESC possess features of mesenchymal cells and an impaired differentiation potential. They next identify genetic alterations which might explain the proliferative and differentiation phenotype of their variant. They also observe that vhESC possesses a higher level of methyltransferases (MT) compared to wt hESC and that NSD3 KO restores epithelial characteristics of vhESC and a normal proliferative rate. They therefore decide to focus the rest of their study on the protein-protein interactions and relevance of MT during transformation using a range of biochemistry approaches.

The authors conclude that:

1. EHMT1/2 interact with NSD3.
2. EHMT1/2 methylate NSD3 on Lysine residue 477 (K477)
3. NSD3 K477 methylation protects NSD3 from proteasome degradation
4. Elevated NSD3 mediates transformation.

Overall this study could in principle provide novel useful information that should be of interest to the audience of the journal. However, I have concerns regarding the quality of the data, their interpretation and the general presentation and clarity of the manuscript. I do not recommend publication at this stage and would advise major revisions.

Major comments:

- 1) *Fig1: The authors claim that their results "demonstrate that V-H9-hESCs display aberrant differentiation features, particularly toward the astrocytic lineage" but the data is weak: Only one image of a GFAP staining is shown where some signal is clearly visible in the variant cell line. The difference in background intensity may indicate different contrast adjustments between the two conditions. To validate this result, the authors could perform qPCR for relevant markers as they did for the cardiac lineage. Furthermore, it would be interesting to determine if the variant cell line has lost competence to differentiate to the astrocytic lineage or if the differentiation is only delayed / less efficient? Note that there is no indication of the number of days at which the analysis was performed. It would be helpful to include a small diagram depicting the timeline of the treatments and analysis in the figure. If this differentiation phenotype is confirmed, how do the authors explain the phenotype? Is this phenotype rescued with V-H9-NSD3 KO cells?*

Response: For the image presented in the manuscript, **Fig.1.H** (Fig. 2C in revised manuscript) we had adjusted the background for both H9-hESC and V-H9-hESC derived glial cells. This was done to catch any faint signal of GFAP in V-H9 hESCs. We have provided original and brightness adjusted images for the reviewer to go through in **Ref Fig1 below**.

Figure Legend for Ref Fig.1: Immunostaining for GFAP in H9-hESCs and V-H9-hESCs. (Scale bar= 100 μ m).

To demonstrate the aberrant astroglial potential of V-H9 hESC we evaluated GFAP protein expression by immunofluorescence. Glial fibrillary acidic protein (GFAP) protein is a reliable marker of astrocyte differentiation and maturation and has been extensively used to identify the astrocytic lineage. Our studies identified diminished expression of GFAP in V-H9-hESC derived astroglia. As per reviewers suggestion we have also performed qRT-PCRs for key markers involved in NSC and astroglial differentiation. Transcription factor PAX6 is expressed in neural stem cells (NSCs) derived from human embryonic stem cells (hESCs), neural stem/progenitor cells (NSCs), astroglial cells and throughout the central nervous system. Pax6 is known to positively regulate neurogenesis via controlling gene regulatory networks (Sansom SN et al, Plos Genet; 2009), and its role as an intrinsic fate determinant of the neurogenic potential of glial cells is also reported (Heins et al Nature Neuroscience 2002). Consistent with the previous reports we observed that the neural stem cells and glial cells (derived from both H9 and V-H9) expressed Pax6 mRNA. We also noticed substantially elevated Pax6 expression V-H9 hESC derived glial cells (Fig. 2F). While our data was consistent for neurogenic potential of V-H9 hESCs, we still did not understand the hampered astroglia differentiation. To address this, we next studied transcription factor Nuclear factor IA (NFIA) which acts as a molecular switch to trigger human glial differentiation (Tchieu et al 2019; Nature Biotechnology) (Li et al; 2018 Stem Cell Reports). Transient expression of NFIA is sufficient to induce glial differentiation of NSCs derived from human pluripotent stem cells within 5 days and to convert these cells into astrocytes (Tchieu et al 2019; Nature Biotechnology) (Li et al; 2018 Stem Cell Reports). Upon assessing the expression of NFIA we identified minimal expression of NFIA in pluripotent stem cells and NSCs (Ct Value-37). Glial expression was

induced in H9-hESC derived glia, however, V-H9 hESC derived glial demonstrated poor induction of NFIA (Fig. 2F). Collectively, these results demonstrate that V-H9-hESCs display aberrant differentiation features particularly toward the astroglial lineage due to weak induction of key transcription factor NFIA. The data is presented in Ref. Fig 2A below.

Next, we tested the contribution of elevated NSD3 in compromised astoglia differentiation from V-H9-hESC. Towards this H9-hESCs, V-H9-hESCs and V-H9-hESC^{NSD3-KD} cells were differentiated into glial cells as described in Sup Fig 2A. We then performed the qRT-PCR analysis for the key transcription regulators Pax6, NFIA and GFAP as surrogate markers for astroglial differentiation. Our results identified that in V-H9-hESC^{NSD3-KD} cells; aberrant expression of these genes was largely rescued confirming the role of NSD3 in astroglial differentiation (Fig 5G). We have included this data in Fig. 2 and Fig. 5 of the revised manuscript. As requested schema is also provided in Supp. Fig 2A. The data is presented in Ref. Fig 2B below.

Figure Legend for Ref Fig.2: **A)** qPCR analysis of Pax6, NFIA and GFAP in H9-hESC & V-H9-hESC, H9-NSCs & V-H9-NSCs and H9-Astroglia & V-H9-Astroglia **B)** qPCR analysis of Pax6, NFIA and GFAP in H9-hESC, V-H9-hESC and V-H9-hESC-NSD3KD derived Astro-glial cells.

Fig 2H could be an interesting source of information to delineate the genetic changes associated with transformation. Only CDH1 and BCL2L2 are annotated and these genes are not the most significantly differentially regulated. Could the authors include the list of genes as a supplementary table and provide at least a gene ontology analysis in their main figure?

Response: We had appended analysed RNA-Seq XL file for differentially regulated genes during manuscript submission. We indicated differential regulation EMT genes such as E-cad, Snail and Zeb1 in the side table which was masked while assembling the figure. We sincerely apologize for this oversight. This is now indicated in Fig. 3D. As per reviewers suggestion we have also performed gene ontology analysis and identified that process such as migration, neuronal differentiation, chromatin organization, RNA Pol 2 mediated transcription and protein ubiquitination are altered in H9 vs. V-H9 hESCs which is consistent with various functional data such as EMT, stability of chromatin modifiers

and aberrant neural differentiation. We have provided this information in **Ref Fig. 3** below and Fig. 3E and Sup Fig. 3G of the revised manuscript.

Figure Legend for Ref Fig.3: Gene ontology analysis of differentially regulated genes in H9-ESCs and V-H9-ESCs.

2) *Figure 3 is particularly important for the main thesis of the paper and is used to demonstrate that EHMT 1 and 2 can methylate NSD3. I am not convinced by the claim that EHMT1 is able to methylate NSD3. Interactions between EHMT1 and NSD3 S and L seem weak (Fig 3B where we see a faint band compared to the very strong signal in the input - Fig3C, the Flag antibody detects multiple bands, how can the authors be sure they detect a specific product - why not use their NSD3 antibody directly) While I agree that a higher NSD3S methylation is detected when EMHT2 is overexpressed, I cannot see the same effect when EMHT1 is overexpressed. Please revise in the main text NB: there is no indication of the number of time this experiment was repeated. Given the importance of the data, I would recommend the authors to show at least 2 replicates.*

Response: In cells, EHMT1 and EHMT2 exist as heteromeric complex (Tachibana et. al 2005). Among these EHMT2 is a predominant methyltransferase, while EHMT1 is known to provide stability EHMT2 protein (Tachibana et. al 2004). To satisfy reviewers concern regarding EHMT1's association with NSD3, we directly immunoprecipitated EHMT1 from V-H9-hESCs and probed for NSD3L. Our results clearly demonstrated the association between EHMT1 and

NSD3L. This data is presented in Fig. 3B of the revised manuscript. Additionally we overexpressed V5-EHMT1 in combination with NSD3L-EGFP or Flag-NSD3S. Immunoprecipitation using V5 antibody showed interaction with both NSD3L and NSD3S. This data is included in Sup Fig. 4 A-B. Taken together EHMT1's interaction with NSD3 was confirmed using endogenous, overexpression of tagged protein and mass spectrometry, the data provided Fig 4B, p Sup Fig. 4A-E of the revised manuscript and **Ref Fig 4**.

Ref Fig. 4

Figure Legend for Ref Fig.4: A) V-H9-hESC lysates were subjected to Immunoprecipitation with anti EHMT1 antibody and the elute was probed for NSD3 and EHMT1. Presence of NSD3 and EHMT1 indicated association between the two proteins in variant cells. B-C) tagged proteins were overexpressed in 293 cells and EHMT1 was IPed using V5 antibody. Elutes were probed with GFP (for NSD3L-GFP) and Flag (Flag-NSD3S). Presence of NSD3L and NSD3S and EHMT1 indicated association between the two proteins.

We agree with reviewer that EHMT2 methylates NSD3 which we were not able to see with EHMT1 overexpression (Fig 4C of the revised manuscript and Ref Fig 5 below). We have indeed repeated this experiment two times. As mentioned by reviewer, this is key data of the manuscript and therefore we confirmed EHMT2 mediated methylation of NSD3 by overexpressing tagged proteins (NSD3L-GFP along F1.EHMT2 or F1.NSD3S along with EHMT2-GFP) which was presented in SupFig.3D&E in earlier version (Fig 4D, E of the revised manuscript). We will be happy to provide the full blots/raw data for these images.

Figure Legends for Ref Fig 5: A) 293 cells were transfected with indicated plasmids and immunoprecipitated with methylated lysine antibody and immunoblotted for indicated antibodies. B) 293s was transfected with indicated plasmids and immunoprecipitated with GFP antibody and immunoblotted for pan methylated lysine antibody and the same blot was re-probed for total protein using GFP antibody C) 293s was transfected with indicated plasmids and immunoprecipitated with flag antibody and immunoblotted for pan methylated lysine antibody and the same blot was re-probed for total protein using flag antibody.

Our results also demonstrated that EHMT2 overexpression is correlated with increased NSD3L and NSD3S methylation. Consistent with methylation data overexpression of EHMT2 but not EHMT1 increased NSD3 protein levels. We had provided EHMT2 overexpression data in the manuscript Fig 7A (Fig. 8A of revised manuscript) and left out EHMT1 overexpression data. We have now provided this data in **Ref Fig 6** and included Sup Fig. 8B of the revised version of the manuscript.

In *in vitro* assays such as luminescence based methylation kinetics and NMR studies we have observed the methylation of NSD3 by EHMT1 and EHMT2. Therefore we had generalized the statement. Reviewers point is well taken and we explained this difference and amended the statement in the result section (Result - 4) of the revised manuscript.

Figure Legend for Ref Fig. 6: A) Western blot analysis of endogenous NSD3L & NSD3S levels in 293s upon overexpression of Flag-EHMT2 B) Quantitation of NSD3 L & S proteins upon overexpression of EHMT2 in 293s

Fig 3E raises concerns. The image intensity is clipped and gives the impression that information is being hidden. We observe signal in the absence of SAM which is odd. We also observe signal in the absence of EHMT2_SET. Is this an indication that NSD3 may methylate itself, or a technical artifact? There also seems to be a plateau effect (no further increase in methylation beyond 0.5x EHMT2). Please comment on all these points.

Response: We apologize for the image quality. We have provided original full blots of data presented in **Ref. Fig 7** for the reviewer to go through. We have seen increased methylation of NSD3 in dose dependent manner in 0.5 and 1X concentration of EHMT2_SET (lanes 3 &4). NSD3 methylation signal was dampened when SAM was not added in the reaction mix (lane5) indicating the specificity of the reaction. Pan methyl antibody, while it works clean for cell lysates, we observed that there is non-specific binding to small extent with pure proteins. Methylation of NSD3 by EHMT2 is the key data of the manuscript and therefore we have confirmed this claim not only by in vitro methylation assay, but also by using several other in vitro and in vivo using assays such 1. Fluorescence based methylation assay, 2. NMR studies, 3. IP MeK followed by IB NSD3 upon over expression of EHMT2, 4. endogenous mono-MeK, di-MeK and tri MeK IPs and probing for NSD3, 5. identification of methylation site and 6. generating methylation deficient mutants. Given the data was validated through multiple experiments, we hope that reviewers concern is addressed.

**Ref Fig. 7**

**Figure Legend for Ref Fig. 7:** Purified 8xHis-NSD3 was incubated overnight at 37°C along with SAM (S-adenosyl-L-Methionine) and increasing concentration of purified EHMT2-SET domain. Reaction mixture was immunoblotted for the indicated antibodies.

*Fig 3F seems to be mis-annotated. there are 6 lanes in the image but only 5 are annotated. See the negative bands between Input and IP-IgG. Overall this figure does not inspire confidence and I would recommend that the authors provide raw images with their revision.*

**Response:** We apologize for the labelling oversight. The lane showing negative bands between Input and IP-IgG is of molecular weight marker. We have now provided lower exposure of the original blot with the marker overlay (**Ref. Fig 8**) to show the lane indeed represents the marker. Hope this inspire reviewers confidence.

**Ref Fig. 8**

Figure Legend for Ref Fig. 8: Immunoprecipitation with anti-methylated lysine specific antibodies and immunoblotted for NSD3 from H9-hESCs and V-H9-hESCs.

- 3) *Fig 5A. The authors claim here that they could detect "a strong interaction of EHMT2 with*
- 4) *of EHMT2 with NSD3 N-term (1-173) domain". However, there are multiple concerning issues with this panel:*
 - *There is no condition probing for EHMT2 and NSD3S interaction in this experiment (lane 7 is only showing that the flag antibody can bind to Flag-NSD3S)*
 - *The image shows lanes with different contrast adjustments. This is not acceptable as this does not allow us to compare lanes with one another. Lane 8 for example has been contrast adjusted but the corresponding input was not. Even if a band is detected, we cannot conclude that there is a strong interaction here.*
 - *Even more problematic, the input for NSD3 1-173 was contrast adjusted to show a very faint band indicating that very little protein was produced in this condition. On the other hand, the corresponding flag pull-down was not adjusted, thus "conveniently" showing no band at all. If the authors wish to maintain their claim, they should repeat this experiment and provide original raw images of the blots.*

Response:

Upon overexpression of Flag-NSD3S, NSD3(1-384) and NSD3 (1-173) followed by Immunoblotting with NSD3 specific antibody we noticed that they do not express in equal amount. Specifically, NSD3 (1-173) is expressed at comparatively low levels and therefore it has become difficult to get all of them developed at the same time upon immunoblotting. Therefore we have done independent experiments. Tagged EHMT2 was co-expressed with tagged NSD3 proteins of containing various domains. Immunoprecipitation using tag specific antibody bound to EHMT1 followed by immunoblotting for NSD3 identified full length, short form and NSD3(1-384) interact with EHMT2. This data is provided below in **Ref Fig 9**. Hope this addresses reviewers concern.

Figure Legend for Ref Fig. 9: A-D) Co-immunoprecipitation of Flag EHMT2 followed by immunoblotting with NSD3L ,NSD3S shows NSD3 (1-384) or NSD3 (1-173) domain with EHMT2.

5. Fig 6A and B present an *in vitro* assay measuring the methylation of a 21 aa peptide by either EHMT1 or EHMT2. I am a little confused by the approach as the authors mentioned earlier that the PWWP domain was essential for NSD3 and EHMT2 interaction, and therefore methylation. The authors should clarify their rationale and incorporate the sequence and position of the 21 aa peptide in the main figure. (The fact that the peptide contains other lysine residues is an important information for interpretation of the data)

Response: Protein-Protein Interactions (PPIs) in the cellular milieu occurs in a complex mixture of biomolecules. Domains such as PWWP facilitate the enzyme-substrate interaction in several ways in such complex settings. Experiments such as Co-IP confirms interactions using a whole cell extract where proteins are present in their native form in a complex mixture of cellular components and therefore, we can study importance of the domains necessary for NSD3 and EHMT2 interactions.

Contrary, Fig 6A and B (Fig 7A and B of the revised manuscript) represents an *in vitro* assay where in enzyme and substrate are present in the surplus in the tube and devoid of other complexities/competitions, the reaction proceeds irrespective of these binding domains.

While looking for the possible methylation site we observed Histone H3K9-like “ARKS” motif in the NSD3 protein. Hence we designed the 21 aa peptide containing the ARKS motif at the centre, the peptide sequence is “VKIAWKTAAAARKSLPASITM”. There are three lysine residues in the peptide. The lysine methylation in the “ARKS” motif is confirmed by the mutation of lysine to alanine in the NSD3 K477A mutant peptide

“VKIAWKTAARASLPASITM”. Since the methylation signal of the mutant peptide was as much as the blank reaction, we concluded that K477 is the lysine that gets methylated by EHMT1 and 2 SET domain. Overall peptide based experiment was designed to study and confirm the lysine that we hypothesised to get methylated (Fig. 7 & Sup Fig. 7). As such peptide based PTM studies have been done in past for histone and non-histone proteins extensively (Rowe EM and Biggar KK 2018; MethodsX). We hope that this information clarifies reviewers concern.

6) Fig 7: The authors make the strong claim that NSD3 K477 methylation protects NSD3 from proteasome degradation. However they do not provide evidence for this. Fig 7E only shows that the addition of the proteasome inhibitor MG 132 leads to a higher level of NSD3 wt and K477R. This only shows that NSD3 is susceptible to proteasome degradation independently of K477. We do not observe any further reduction in the level of K477R compared to the wt in the absence of MG132. Please moderate the claim or perhaps try an anti-ubiquitin antibody against both versions of NSD3 after IP.

Response: We thank the reviewer for this comment. As suggested by reviewer we investigated in detail that the K477 moiety which gets methylated indeed promotes stabilization of NSD3 and antagonizes degradation via ubiquitin proteasome system. Towards this we transfected Flag tagged NSD3S Wt or K477R mutant IRES GFP plasmids in 293 cells. Given this is a transient expression system, transfection efficiency control for Wt. and mutant was based on internal GFP IRES expression (Fig. 8E; left panel). Gapdh served as a total protein loading control (Fig. 8E; left panel). Probing of input lysates for total ubiquitin levels revealed higher levels of ubiquitin signal in MG132 compared to wt. or mutant expressing constructs alone (Fig. 8E; left panel). This confirmed the inhibition of ubiquitin proteasome system mediated degradation of proteins upon MG132 treatment (Fig. 8E; left panel).

Next, 500mg of cell lysate was subjected to Immunoprecipitation using anti Flag antibody and IPed material was probed using anti Flag and poly ubiquitin antibodies. Equal amount of material was IPed in NSD3S Wt, NSD3S K477R and NSD3S K477R + MG132 treated cells (Fig. 8E; right panel). We noticed albeit higher levels of IP flag in NSD3 Wt treated with MG132 lysates (Fig. 8E; right panel). Anti ubiquitin blot demonstrated that the ubiquitin signal was higher in NSD3S mutant lysate compared to NSD3S Wt (Fig. 8E; right panel). This data was consistent with the information in Fig. 7H wherein we observed that NSD3S K477R mutant was unstable. Enhanced ubiquitination signal in NSD3S K477R vs. NSD3S K477R + MG132 IPed material further attested that NSD3S protein undergoes degradation via ubiquitin proteasome machinery (Fig. 8E; right panel).

We also tested if EHMT2 indeed methylates NSD3 at K477 residue and prevents it from ubiquitin mediated degradation. Towards this transfected Flag tagged NSD3S or K477R IRES GFP constructs along with various concentrations of Flag-EHMT2. Lysates were probed for transfection efficiency of

NSD3S or K477R using anti GFP antibody and EHMT2 overexpression monitored by anti-Flag antibody. While there was higher transfection efficiency in single plasmid (NSD3S Wt. or NSD3S mutant) transfected cells (Fig. 8F; left panel)), equal levels of transfection efficiency were achieved in all Flag-EHMT2 co-transfected cells. Dose dependent increasing expression of Flag EHMT2 (140 Kda) was seen exclusively in co-transfected cells (Fig. 8F; left panel)).

Increased expression of EHMT2 increased Flag-NSD3S Wt protein levels (90 Kda) (Fig. 8F; left panel)), however such an effect was not seen in Flag-NSD3S K477R mutant transfected cells (Fig. 8F; right panel)). These results convincingly demonstrate EHMT2 methylated K477R residue on NSD3 and prevents ubiquitin mediated degradation.

Methylation of a lysine residue can preclude the addition of a ubiquitin moiety on the same lysine residue and increase protein stability. To address if K477 is the residue which is targeted for both methylation and ubiquitination, we performed an experiment, wherein Flag-NSD3S or Flag-NSD3S K477R were overexpressed and supplemented with MG132. Increase in p21 protein upon MG132 addition served as a positive control (Sup Fig. 8H). Probing for anti-Flag antibody demonstrated higher levels of NSD3S in Flag-NSD3S or Flag-NSD3S K477R indicative of stabilization of the NSD3 upon inhibition of ubiquitination proteosome system (Sup Fig. 8H). Since we observed recovery of NSD3 mutant protein only upon MG132 treatment indicated that K477 is the site of methylation but not shared for ubiquitination.

Overall, the data from Fig. 5 (of the revised manuscript) demonstrate the consequence of NSD3 stabilization leading to pluripotent stem cell transformation and in Fig. 8 (of the revised manuscript) we provide understanding on mechanisms by which protooncogene NSD3 levels are kept in check by ubiquitin proteosome system in normal H9 hESCs.

Ref Fig. 10

Figure Legend for Ref Fig. 10: A) Western blot for Immunoprecipitation of 293s transfected with NSD3S Flag and NSD3SK477R treated with and without MG132. The lysates were immunoprecipitated with anti-flag antibody and immunoblotted for flag and Ubiquitin antibodies. GFP was used as a transfection control. GAPDH was used as protein control. **B)** Western blot for NSD3S Flag and NSD3K477R transfected with EHMT2-GFP in a dose-dependent manner in 293s. Lysates were immunoblotted for Flag, GFP (EHMT2) antibodies. Gapdh was used as protein control. GFP (IRES) was used as a transfection control.

Other comments:

* *Generally, the quality of the immunoblots is questionable. For example, we see at least two main bands in Fig 1D but this is not commented on. Same for Fig 3 C with multiple bands. (IP-v5 - IB: Flag); Fig 4D, the lanes are not well aligned with rows including 3 lanes while others include 2 lanes.*

Response: We have provided the original data wherever reviewers have explicitly raised the concerns. We will be glad to upload all the original data for the manuscript.

* *The molecular weight is often missing from the various immunoblots - the ladder is never annotated.*

Response: Molecular weight markers are now indicated in the revised manuscript and ladder is annotated.

* *Most scale bars are difficult to see (Fig 1A, E - absent in I - Fig 4A,E). The scale bar is given in pixel in F-H*

Response: We have mentioned the scale bars in the Figure legends for the clarity.

* *Fig 2H: The legend appears behind the volcano plot and cannot be read.*

Response: Error has been fixed in the revised manuscript.

* *fig legends are incomplete - for example Fig 1F-J, what is the time of analysis post induction of differentiation? Number of independent experiments.*

Response: Details have been provided in the result section of Fig. 2 and the figure legends.

Fig 3E: the acronyms SAM and SET are not defined; Fig 6C and D, what do the pink and blue color represent...

Response: Acronyms have been defined in the revised manuscript. The Pink color in NMR Spectra shows the proton-carbon correlations in –CH– and –CH₃ groups while the blue color shows the proton-carbon correlations in –CH₂– groups.

* *Fig 1E: There is no comment in the main text regarding the Snail staining in Fig 1E. Furthermore both wt and vhes are positive for Snail, please comment*

Response: We apologise for the oversight. Explanation regarding the snail staining is included in the revised manuscript.

** P7 - "Distinct chromosomal abnormalities detected in multiple studies including ours suggest that these cannot be used as specific markers of H9-hESC transformation." I am not sure I understand what the authors mean here - Please clarify*

Response: Number of variant lines have been characterized world-wide and reported to have overlapping and distinct chromosomal alterations. However till data no single genomic marker has been known to be commonly/ consistently altered in all these lines. While we also identified genetic anomalies, we intend to identify non-genetic independent molecular marker that can predict hESC transformation and therefore we focused our analysis on epigenetic regulators. This is now clarified in the revised manuscript.

Reviewer #2

In this study, Rampalli et al. investigated the difference between the commonly used human embryonic stem cell line, H9-hESC, and the variant H9-hESC (V-H9-hESC). The V-H9-hESC exhibited dysregulated EMT-specific markers and compromised differentiation. They focused on the epigenetic level changes and found that V-H9-hESC has increased expression levels in EHMT1, EHMT2, and NSD3. They also found that the NSD3 knockout can rescue the EMT marker dysregulation and the phenotypes in V-H9-hESC. To investigate the mechanism further, they found that EHMT1 and EHMT2 can interact with the PWWP domain in NSD3. They performed IP and methylation assay and found that the K477 residue in NSD2 can be methylated and stabilized in the protein from proteasome degradation. Overall, this study provides an interesting hESC line as an EMT study tool and molecular mechanism for the NSD3 lysine residue methylation from the EHMTs. However, there are several major concerns about the data quality, and the experimental result didn't justify the conclusions well. Here are the specific comments and suggestions:

Major comment:

1. In Figure 1, the author showed the morphology changes in the variant H9-hESC line and changed expression level of EMT-related factor compared to H9-hESC. It is critical to quantify or measure the EMT phenotype using a wound healing assay in variant H9-hESC. It is also acquired in NSD3 KO V-H9-hESC in Fig 4 to check the rescue phenotypes.

Response: Thank you for your comment. As suggested we have performed wound healing assay using H9-hESC and V-H9-hESC and examined migration competency of the V-H9-hESC in response to the mechanical scratch wound. Scratch wounds were created in the colonies of H9 and V-H9-hESC and re-growth into a scratched area was monitored microscopy over three days. In, H9 hESCs scratched area cells migrated as sheet and closed scratch in 72h (Fig. 1F). Whereas in V-H9-hESC we observed single

cells in the scratch area, these cells grew rapidly and occupied the scratched area in 24h (Fig 1G). The scratch area became confluent by 36h. We also quantified the percentage of open wound area which clearly demonstrated that V-H9-hESC possess accelerated cell migration properties (Fig. 1H). Overall, our results V-H9-hESC demonstrated EMT and enhanced migration.

We also tested migration of V-H9-hESCs^{NSD3-KD} compared to H9-hESCs and V-H9-hESCs in scratch assay. As expected, V-H9-hESCs migrated and rapidly proliferated in the scratched area in 24h, Scratched area was then confluent by 36 h. However, V-H9-hESCs^{NSD3-KD} closed the wounds at 48h which is albeit slower than V-H9-hESCs and were faster than H9-hESCs (Fig 5F). Interestingly the pattern of V-H9-hESCs^{NSD3-KD} migration resembled H9-hESCs i.e. in the form of sheet. Proliferation and migration data indicated that these properties were partially reverted upon NSD3 knockdown in V-H9-hESC. This information is included in revised manuscript and in Ref Fig below.

Ref Fig. 11

Figure Legend for Ref Fig 11: A) Scratch wound healing of H9-hESCs B) Scratch wound healing of V-H9-hESCs C) Scratch wound healing of V-H9-hESCs-NSD3KD

In RNA-Seq analysis we identified differential regulation EMT genes such as E-cad, Snail and Zeb in H9-hESC, vs V- H9-hESCs. We have now performed gene ontology analysis and identified that process

such as migration, neuronal differentiation, chromatin organization, RNA Pol 2 mediated transcription and protein ubiquitination are altered in H9 vs. V-H9 hESCs which is consistent with various functional data such as EMT, stability of chromatin modifiers and aberrant neural differentiation. We have provided this information in **Ref Fig 3** for reviewers to go through and will be happy to include in the revised manuscript. Together these data will strengthen our claim of EMT phenotype observed in V-H9-hESC.

Figure Legend for Ref Fig.3: Gene ontology analysis of differentially regulated genes in H9-ESCs and V-H9-ESCs.

2.H3K36me2 is significantly increased in the V-H9-hESC line compared to the parental line, however, the level of their methyltransferase NSD3L seems mildly increased in the V-H9-hESC. Is there another H3K36 methyltransferase level changes?

Response: Mono- and di-methylation of H3K36 deposited by NSD proteins (NSD1, NSD2 and NSD3) plays an important role in maintenance of chromatin integrity and regulate the expression of genes that control cell division, apoptosis, DNA repair, and epithelial–mesenchymal transition (EMT). Mutations or aberrant expression of NSD family is associated with developmental defects and cancers. Particularly, NSD2 and NSD3 are frequently overexpressed in various cancers.⁴⁴ To address reviewers concern we have performed western blotting for other two NSD family members (NSD1 and NSD2) that are known to catalyse H3K36 methylation. Interestingly, all three NSD family members were upregulated was enriched in V-H9-hESCs (Fig. 3B,C) compared to H9-hESCs. While we noticed very low levels of NSD1 and NSD3 (both long and short forms) in H9

hESCs, NSD2 expression was undetectable in H9 hESCs (Fig. 3B, Sup Fig. 3C). Investigation into the levels of NSD1 and NSD2 in V-H9-hESCs^{NSD3-KD} revealed that NSD1 protein was reduced in these cells, however NSD2 levels remained unchanged (Sup Fig. 5G). Persistent higher growth rates in VhESC^{NSD3-KO} despite reduction of NSD3 and c-Myc indicates that this can be attributed to several factors including elevated NSD2 in V-H9-hESCs. This data is presented in revised manuscript and in **Ref Fig.12** below.

Figure Legend for Ref Fig.12. Western blot analysis of NSD1 & NSD2 in H9-hESCs, V-H9-hESCs and V-H9-hESCs-NSD3KD

3. In Figure 2, Panel AB: The author found that some genes that are important for cell proliferation and differentiation have small amplifications and deletions on the genome of the V-H9-hESC line. How do they preclude the possibility that the issue causes the EMT phenotype in the V-H9-hESC line?

Response: Thank you for commenting regarding this important aspect. As shown in Fig 4 (Fig. 5 of revised manuscript) NSD3 knockdown was performed in V-H9-hESC to generate NSD3 KD V-H9-hESC line. NSD3 KD V-H9-hES cells share all the small amplifications and deletions that are present on the genome of V-H9-hESCs. Given that genomic background was similar in both cell lines and yet NSD3 KD reversed the EMT phenotype and cellular proliferation we concluded the role of NSD3 stabilization in EMT phenotype. Consistent with that overexpression of NSD3 in wild-type cells led to the upregulation of mesenchymal markers and enhanced proliferation. V-H9-hESC is a randomly evolved line in the culture and the possibility of small amplifications and

deletions causing EMT during initial phases of evolution cannot be precluded, however these phenotypes in V-H9-hESC now seem to be dependent on NSD3 overexpression.

We thought about this valuable comment from reviewers very carefully and revisited our data. We notice that proliferation, migration and differentiation defects were not completely reversed in NSD3 KD V-H9-hESC and were partially rescued. As reviewer pointed out correctly these do not allow us to preclude the role of genetic mutation and the impact of other epigenetic regulators such as NSD2 in defects mentioned above. Additionally several of the mutations in V-H9-hESCs may have a role in differentiated derivatives and therefore impact the differentiation process. For example a mutation in TNNT2 gene impacts cardiac beating which is correlated in our cardiac differentiation experiment. We have mentioned these details in the discussion section of the revised manuscript.

4. In Figure 5, Panel A: The immunoblotting data are spliced, the authors need to provide the original data (also in Figure 3 A). It is confusing why there is a Flag-NSD3S-only group and the pull-down efficiency is low in the NSD 1-384 group. It is worth replacing the Flag-NSD3S-only group with NSD3S (without Flag) and Flag-EHMT2 co-transfection group.

Response: Upon overexpression of Flag-NSD3S, NSD3(1-384) and NSD3 (1-173) followed by immunoblotting with NSD3 specific antibody we noticed that they do not express in equal amount. Specifically, NSD3 (1-173) is expressed at comparatively low levels and therefore it has become difficult to get all of them developed at the same time upon immunoblotting. Therefore we have done independent experiments. Tagged EHMT2 was co-expressed with tagged NSD3 proteins of various domains. Immunoprecipitation using tag specific antibody bound to EHMT2 followed by immunoblotting for NSD3 identified full length, short form and NSD3(1-384) interact with EHMT2. This data is provided below in **Ref Fig 9**. Hope this addresses reviewer's concern.

Figure Legend for Ref Fig. 9: A-D) Co-immunoprecipitation of Flag EHMT2 followed by immunoblotting with NSD3L ,NSD3S shows NSD3 (1-384) or NSD3 (1-173) domain with EHMT2.

5. The authors investigated the stabilization of NSD3 as being associated with the methylation of K477 residue and proteasome degradation. Although they found the K477 residue didn't share the ubiquitination site, they still suggested that the NSD3 protein methylation prevented the ubiquitination in the text and Figure 8. The data for this part is insufficient.

Response: We thank the reviewer for this comment. As suggested by reviewer we investigated in detail that the K477 moiety which gets methylated indeed promotes stabilization of NSD3 and antagonizes degradation via ubiquitin proteasome system. Towards this we transfected Flag tagged NSD3S Wt or K477R mutant IRES GFP plasmids in 293 cells. Given this is a transient expression system, transfection efficiency control for Wt. and mutant was based on internal GFP IRES expression (Fig. 8E; left panel). Gapdh served as a total protein loading control (Fig. 8E; left panel). Probing of input lysates for total ubiquitin levels revealed higher levels of ubiquitin signal in MG132 compared to wt. or mutant expressing constructs alone (Fig. 8E; left panel). This confirmed the inhibition of ubiquitin proteasome system mediated degradation of proteins upon MG132 treatment (Fig. 8E; left panel).

Next, 500mg of cell lysate was subjected to Immunoprecipitation using anti Flag antibody and IPed material was probed using anti Flag and poly ubiquitin antibodies. Equal amount of material was IPed in NSD3S Wt, NSD3S K477R and NSD3S K477R + MG132 treated cells (Fig. 8E; right panel). We noticed albeit higher levels of IP flag in NSD3 Wt treated with MG132 lysates (Fig. 8E; right panel). Anti ubiquitin blot demonstrated that the ubiquitin signal was higher in NSD3S mutant lysate compared to NSD3S Wt (Fig. 8E; right panel). This data was consistent with the information in Fig. 7H wherein we observed that NSD3S K477R mutant was unstable. Enhanced ubiquitination signal in NSD3S

K477R vs. NSD3S K477R + MG132 IPed material further attested that NSD3S protein undergoes degradation via ubiquitin proteasome machinery (Fig. 8E; right panel).

We also tested if EHMT2 indeed methylates NSD3 at K477 residue and prevents it from ubiquitin mediated degradation. Towards this transfected Flag tagged NSD3S or K477R IRES GFP constructs along with various concentrations of Flag-EHMT2 or EHMT2-GFP. Lysates were probed for transfection efficiency of NSD3S or K477R using anti GFP antibody and EHMT2 overexpression monitored by anti-Flag/anti-GFP antibody. While there was higher transfection efficiency in single plasmid (NSD3S Wt. or NSD3S mutant) transfected cells (Fig. 8F; left panel), equal levels of transfection efficiency were achieved in all Flag-EHMT2/ EHMT2-GFP co-transfected cells. Dose dependent increasing expression of EHMT2 (140 Kda) was seen exclusively in co-transfected cells (Fig. 8F; left and right panel)).

We noticed that increased expression of EHMT2 increased Flag-NSD3S Wt protein levels (90 Kda) (Fig. 8F; left panel), however such an effect was not seen in Flag-NSD3S K477R mutant transfected cells (Fig. 8F; right panel)). These results convincingly demonstrate EHMT2 methylated K477R residue on NSD3 and prevents ubiquitin mediated degradation.

Methylation of a lysine residue can preclude the addition of a ubiquitin moiety on the same lysine residue and increase protein stability. To address if K477 is the residue which is targeted for both methylation and ubiquitination, we performed an experiment, wherein Flag-NSD3S or Flag-NSD3S K477R were overexpressed and supplemented with MG132. Increase in p21 protein upon MG132 addition served as a positive control (Sup Fig. 8H). Probing for anti-Flag antibody demonstrated higher levels of NSD3S in Flag-NSD3S or Flag-NSD3S K477R indicative of stabilization of the NSD3 upon inhibition of ubiquitination proteasome system (Sup Fig. 8H). Since we observed recovery of NSD3 mutant protein only upon MG132 treatment indicated that K477 is the site of methylation but not shared for ubiquitination. The data is presented below in Ref. Fig 10.

Overall, the data from Fig. 5 (of the revised manuscript) demonstrate the consequence of NSD3 stabilization leading to pluripotent stem cell transformation and in Fig. 8 (of the revised manuscript) we provide understanding on mechanisms by which protooncogene NSD3 levels are kept in check by ubiquitin proteasome system in normal H9 hESCs.

Ref Fig. 10

Figure Legend for Ref Fig. 10: A) Western blot for Immunoprecipitation of 293s transfected with NSD3S Flag and NSD3SK477R treated with and without MG132. The lysates were immunoprecipitated with anti-flag antibody and immunoblotted for flag and Ubiquitin antibodies. GFP was used as a transfection control. GAPDH was used as protein control. B) Western blot for NSD3S Flag and NSD3K477R transfected with EHMT2-GFP in a dose-dependent manner in 293s. Lysates were immunoblotted for Flag, GFP (EHMT2) antibodies. Gapdh was used as protein control. GFP (IRES) was used as a transfection control.

6. Some sentences didn't align the data or need more explanation. As in P.7, "Distinct chromosomal abnormalities detected in multiple studies including ours suggest that these cannot be used as specific markers of H9-hESCs transformation." There is no evidence to support this phenomenon. Some sentence also seems redundant. The author needs to modify the description to be more precise in this manuscript.

Response: Number of variant lines have been characterized world-wide and reported to have overlapping and distinct chromosomal alterations. However till data no single genomic marker has been known to be commonly/ consistently altered in all these lines. While we also identified genetic anomalies, our intention was to identify non-genetic molecular marker that can predict hESC transformation and therefore we focused our analysis on epigenetic regulators. This is now clarified in the revised manuscript.

Minor comment:

1. *It needs better image quality (some have shallow resolution), and the figure arrangement (is not consistent) in this manuscript.*

Response: We have provided higher resolution images and aligned the figures consistently.

2. *Better characterization of cell differentiation in Fig1. For example, in Fig1 I, J didn't show the differentiation ability of V-H9-hESC to mesoderm precursor cells.*

Response: We identified cystic IB upon cardiac differentiation and expression of early mesodermal precursor markers such as TBX6 and Brachyury expression is also tested.

3. *Fig 2 G, H figure issue, and the correlation with epigenetic changes (seems not related to other data)*

Response: Apologies for the oversight we have fixed the issue. We have performed gene ontology analysis and identified that process such as chromatin organization and RNA Pol 2 mediated transcription are altered in H9 vs. V-H9 hESCs.

4. *NSD3 KO/KD issue: The knockout cell line also expresses NSD3L and NSD3S (methods : bulk or single colony)*

Response: We have now indicated these details in the method section.

5. *What is the FT group in Fig 5 B?*

Response: FT refers to Flow through will be indicated in the revised manuscript.

6. *OE NSD3 didn't affect EHMT2 level, however, KD NSD3 reduced EHMT2 level*

Response: We believe that EHMT2 is influenced in KD NSD3 via indirect mechanism which we cannot describe with the available information. As such we have mentioned in discussion that regulation of EHMT2 requires separate investigation.

7. *In Figure 7, Panel F: The protein stability assay didn't reflect the correlation of NSD3 stabilization and the increased cMyc level.*

Response: The enhanced levels of NSD3 are known to stabilize c-Myc protein via protein-protein interaction in cancer cells. In Fig 1 we have shown that V-H9-hESC expresses low c-Myc transcripts compared to H9-hESCs, however, we noticed higher c-Myc protein in variants. Therefore, we believe that this could be due to stabilization of c-Myc protein via NSD3. Currently we do not know the exact relation between extent of NSD3 upregulation and c-Myc

stabilization in V-H9 hESC, however we did notice enhanced c-Myc levels at 6h time point (Fig 8D).

November 20, 2024

RE: Life Science Alliance Manuscript #LSA-2024-02871-TR-A

Dr. Shravanti Rampalli
Institute of Genomics and Integrative Biology
DTC Bus Depot, South Campus, Mathura Road, near to Sukhdev Vihar, New Delhi
New Delhi, Delhi 110025
India

Dear Dr. Rampalli,

Thank you for submitting your revised manuscript entitled "NSD3 protein methylation and stabilization transforms human ES cells into variant state". We would be happy to publish your paper in Life Science Alliance pending final revisions necessary to meet our formatting guidelines.

- please address the Reviewer's remaining comments
- please be sure that the authorship listing and order is correct
- please upload your main manuscript text as an editable doc file;
- please upload all figure files as individual ones, including the supplementary figure files; all figure legends should only appear in the main manuscript file
- please add your main and supplementary figure legends to the main manuscript text after the references section
- you may want to consider uploading Figure 9 as a Graphical Abstract rather than as a figure, but this is up to you
- the supplemental text should only contain the sequencing reads, and be mentioned in the main text with a callout

FIGURE CHECKS:

- please explain the black line in Figures 4A, 6C and 7E in their respective figure legends

LSA now encourages authors to provide a 30-60 second video where the study is briefly explained. We will use these videos on social media to promote the published paper and the presenting author (for examples, see <https://docs.google.com/document/d/1-UWCfbE4pGcDdcgzcmiuJl2XMBJnxKYeqRvLLrLS08s/edit?usp=sharing>). Corresponding or first-authors are welcome to submit the video. Please submit only one video per manuscript. The video can be emailed to contact@life-science-alliance.org

A. FINAL FILES:

B. MANUSCRIPT ORGANIZATION AND FORMATTING:

Sincerely,

Reviewer #1 (Comments to the Authors (Required)):

The authors made substantial changes to their manuscript and provided satisfactory answers to my initial concerns. Overall I think this has strengthen the manuscript and I can now recommend publication with minor revisions.
Congrats to the authors for their efforts.

Minor points:

* The legend in Fig 2 is incomplete: Please indicate the number of replicates for qPCR data, what the error bars indicate, state which statistical test was used and what the asterisks mean. NB: Statistical tests should be performed between H9 and v-H9 for each condition (hESC,NSC,Astroglia) in order to test the hypothesis that a difference exists between the two cell lines.

* The same comment can be made for the qPCR data in Fig 5G.

* In vitro methylation assay - I strongly encourage the authors to include the peptide sequences inside Figure 7 and explain in the main text why the PWWP domain is not necessary for methylation to take place in this in vitro setting.

Point wise response to Reviewers comment

Reviewer #1 (Comments to the Authors (Required)): Minor points:

** The legend in Fig 2 is incomplete: Please indicate the number of replicates for qPCR data, what the error bars indicate, state which statistical test was used and what the asterisks mean. * NB: Statistical tests should be performed between H9 and v-H9 for each condition (hESC, NSC, Astroglia) in order to test the hypothesis that a difference exists between the two cell lines.*

Response: In order to do a direct comparison of gene expression throughout the astroglial differentiation process (from ESC to NSCs and then glial cells), we have grouped the data and used h9 ESCs as the control. As the dataset included more than two conditions, we had to use Ordinary one-way Anova to test the significance. We have performed the Statistical Analysis for H9 and V-H9 for all three biological replicates (n=3) in each of the conditions mentioned in the graph of Figure 2: D-F as suggested by the reviewer. For comparison between H9 vs. V-H9 ESCs in each sperate condition (ESCs, NSCs and glial cells), we performed with unpaired t-Test to test the significance or each of the genes tested. For GFAP, Unpaired t-Test *(p value=0.0165) for H9 and V-H9 ESCs, **(p value= 0.0035) for H9 vs. V-H9 NSCs and *****(p value <0.0001) for H9 and V-H9 astroglial cells. Error bars: s.e.m. For NF1A, Unpaired t-Test ns (p value= 0.0592) for H9 and V-H9 hESCs, **(p value= 0.0067) for H9 vs.V-H9 NSCs and *****(p value <0.0001) for H9 and V-H9 astroglial cells. Error bars: s.e.m. For PAX6, Unpaired t-Test ns (p value=0.0938) for H9 and V-H9 ESCs, ns(p value=0.3761) for H9 vs. V-H9 NSCs and *****(p value <0.0001) for H9 and V-H9 astroglial cells. Error bars: s.e.m. This information has now been included in the revised manuscript under the statistical methods.

** The same comment can be made for the qPCR data in Fig 5G.*

Response: As mentioned in the previous comment response, we have done the statistical analysis for H9 and V-h9 ESCs separately as well as for H9 and V-H9-hESC^{NSD3KD}. This however isn't represented in the figures included in the manuscript.(n=3) .For GFAP, Unpaired t-Test **** (p value= <0.0001) for H9 and V-H9 ESCs, **(p value= 0.0023) for H9 and V-H9-hESC^{NSD3KD} .Error bars: s.e.m. For NF1A, Unpaired t-Test **** (p value= <0.0001) for H9 and V-H9-hESC, **(p value= 0.0085) for h9 vs and V-H9-hESC^{NSD3KD} . Error bars: s.e.m. For PAX6, Unpaired t-Test **(p value=0.0059) for H9 and V-H9 hESCs,

and** (p value= 0.0035) for H9 and V-H9-hESC^{NSD3KD} Error bars: s.e.m. This information has now been included in the revised manuscript under the statistical methods.

** In vitro methylation assay - I strongly encourage the authors to include the peptide sequences inside Figure 7 and explain in the main text why the PWWP domain is not necessary for methylation to take place in this in vitro setting.*

Response: As suggested by reviewer we have included peptide sequence and domain related explanation in the revised manuscript.

- please be sure that the authorship listing and order is correct

Response: Authorship listing is in correct order

- please upload your main manuscript text as an editable doc file

Response: We have uploaded manuscript text in word doc format which is editable.

- please upload all figure files as individual ones, including the supplementary figure files; all figure legends should only appear in the main manuscript file

Response: We have uploaded all figure files including the supplementary figure in individual format.

- please add your main and supplementary figure legends to the main manuscript text after the references section

Response: Main and supplementary figure legends are now added after reference section in revised manuscript.

- you may want to consider uploading Figure 9 as a Graphical Abstract rather than as a figure, but this is up to you

Response: As suggested we have uploaded Figure 9 as graphical abstract.

- the supplemental text should only contain the sequencing reads, and be mentioned in the main text with a callout

Response: The supplemental text now only contain the sequencing reads, and is mentioned in the main text.

** Summary blurb (enter in submission system): A short text summarizing in a single sentence the study (max. 200 characters including spaces). This text is used in conjunction with the titles of papers, hence should be informative and complementary to the title. It should describe the context and significance of the findings for a general readership; it should be written in the present tense and refer to the work in the third person. Author names should not be mentioned.*

Response: The study explores the role of methylation and stabilization of lysine methyltransferase NSD3, an oncogene as a driving factor for transformation of human ESCs from epithelial to mesenchymal state.

November 29, 2024

RE: Life Science Alliance Manuscript #LSA-2024-02871-TRR

Dr. Shravanti Rampalli
Institute of Genomics and Integrative Biology
DTC Bus Depot, South Campus, Mathura Road, near to Sukhdev Vihar, New Delhi
New Delhi, Delhi 110025
India

Dear Dr. Rampalli,

Thank you for submitting your Research Article entitled "NSD3 protein methylation and stabilization transforms human ES cells into variant state". It is a pleasure to let you know that your manuscript is now accepted for publication in Life Science Alliance. Congratulations on this interesting work.

DISTRIBUTION OF MATERIALS:

Again, congratulations on a very nice paper. I hope you found the review process to be constructive and are pleased with how the manuscript was handled editorially. We look forward to future exciting submissions from your lab.

Sincerely,
